# Learning Axis-Aligned Decision Trees with Gradient Descent

## Abstract

Decision Trees are commonly used for many machine learning tasks due to their high interpretability. However, learning a decision tree from data is a difficult optimization problem, since it is non-convex and non-differentiable. Therefore, common approaches learn decision trees using a greedy growth algorithm that minimizes the impurity at each internal node. Unfortunately, this greedy procedure can lead to suboptimal trees.

In this paper, we present a novel approach for learning hard, axis-aligned decision trees with gradient descent. This is achieved by applying backpropagation with a straight-through operator on a dense decision tree representation that jointly optimizes all decision tree parameters. We show that our gradient-based optimization outperforms existing baselines on several binary classification benchmarks and achieves competitive results for multi-class tasks. To the best of our knowledge, this is the first approach that attempts to learn hard, axis-aligned decision trees with gradient descent without restrictions regarding the structure.

## 1 Introduction

Decision trees (DTs) are one of the most popular machine learning models and are still frequently used today. Especially with the increasing interest in explainable artificial intelligence (XAI), DTs regained popularity. However, learning a DT is a difficult optimization problem, since it is non-convex and non-differentiable. Finding an optimal DT for a specific task is a NP-complete problem (Laurent & Rivest, 1976). Therefore, the common approach to construct a DT based on data is a greedy procedure (Breiman et al., 1984; Quinlan, 1993) that minimizes the impurity at each internal node. The algorithms that are still used today, namely CART Breiman et al. (1984) and C4.5 Quinlan (1993), date back until the 1980s and since then remained mostly unchanged. Unfortunately, a greedy algorithm only yields to locally optimal solutions and therefore can lead to suboptimal trees. We illustrate the issues that can arise when learning DTs with a greedy algorithm in the following example:

*Example 1* The Echocardiogram dataset[1] deals with predicting one-year survival of patients after a heart attack based on tabular data from an echocardiogram. Figure 1 shows two decision trees. The left tree is learned using a greedy algorithm (CART) and the right tree is learned using our gradient-based approach. We can observe that the greedy procedure leads to a suboptimal tree with a significantly lower performance. Splitting on the *wall-motion-score* is the locally optimal split (see Figure 1a), but globally, it is beneficial to split based on the *wall-motion-score* with different values conditioned on the *pericarcial-effusion* in the second level (Figure 1b).

The contribution of this paper is a novel approach for learning hard, axis-aligned DTs based on a joint optimization of all parameters with gradient descent, which we call gradient-based decision trees (GDTs). Using a gradient-based optimization, we can overcome the limitations of greedy approaches, as indicated in Figure 1b. We propose a suitable dense DT representation that allows a gradient-based optimization of the tree parameters (Section 3.2). We further present an algorithm that allows us to deal with the non-differentiable nature of DTs, while still allowing an efficient optimization using backpropagation with a straight-through (ST) operator (Section 3.3).

---

[1] Available under: `https://archive.ics.uci.edu/ml/datasets/echocardiogram` (last accessed 16.08.2022)

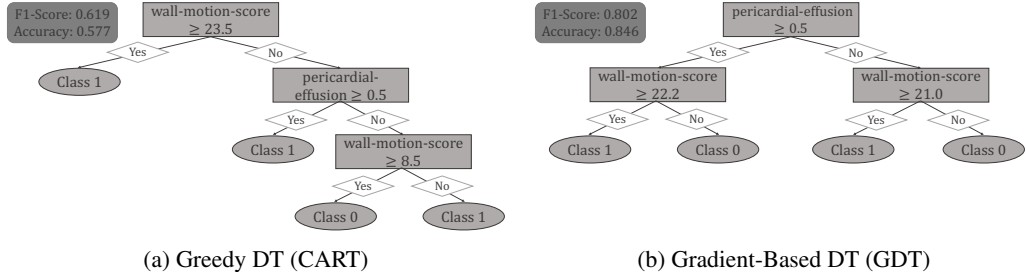

(a) Greedy DT (CART)          (b) Gradient-Based DT (GDT)

Figure 1: **Greedy versus Gradient-Based DT.** Two DTs learned on the Echocardiogram dataset. While the CART DT (left) only makes locally optimal splits, the GDT (right) performs a joint optimization of all parameters, which results in a significantly higher performance.

We empirically evaluate GDTs on several real-world benchmark datasets (Section 4). GDTs outperform existing baselines on several benchmark datasets and achieves competitive results for DT learning. Furthermore, the resulting trees are less prone to overfitting and represent a more accurate quantification of uncertainty, since they comprise probability distributions at the leafs. A gradient-based optimization also provides more flexibility, since splits can be adjusted during the training. This allows an application of GDTs to dynamic scenarios, as for instance online learning.

## 2 RELATED WORK

**Greedy DT Algorithms**  The most prominent and still frequently used algorithms, namely CART (Breiman et al., 1984) and C4.5 (Quinlan, 1993) as an extension of ID3.0 (Quinlan, 1986), data back to 1980s and both follow a greedy procedure to learn a DT. Since then, many variations to those algorithms have been proposed, as for instance C5.0 (Kuhn et al., 2013) and GUIDE (Loh, 2002; 2009). However, until today, none of these algorithms was able to consistently outperform CART and C4.5 as shown by Zharmagambetov et al. (2021).

**Lookahead DTs**  To overcome the issues of a greedy DT induction, many researchers focused on finding an efficient alternative. One solution to mitigate the impact of a greedy procedure are methods that look ahead during the induction (Sarkar et al., 1994). However, Murthy (1996) argue that those methods not only suffer from an enormous increase in the computational complexity, but also suffer from pathology, i.e., they frequently produce worse trees in terms of accuracy, tree size and depth. One explanation could be that lookahead trees, especially without regularization, are prone to overfitting.

**Optimal DTs**  Optimal DTs try to optimize an objective (e.g., the purity) using approximate brute force search to find a globally optimal tree with a certain specification (Zharmagambetov et al., 2021). OCT (Bertsimas & Dunn, 2017) defines the optimization as a mixed integer optimization (MIO) problem which is solved using a MIO solver. In contrast, DL8.5 (Aglin et al., 2020) and GOSDT (Lin et al., 2020) approximate a brute force search using a branch-and-bound algorithm to remove irrelevant parts from the search space. MurTree (Demirović et al., 2022) further uses dynamic programming to reduce the runtime significantly. However, state-of-the-art approaches still require binary data and therefore a discretization of continuous features (Aglin et al., 2020; Demirović et al., 2022; Bertsimas & Dunn, 2017).

**Genetic DTs**  Another way to learn DTs in a non-greedy fashion is using evolutionary algorithms for DT induction. Evolutionary algorithms perform a robust global search in the space of candidate solutions based on the concept of survival of the fittest (Barros et al., 2011). This usually results in a better identification of feature interactions compared to a greedy, local search (Freitas, 2002).

**Oblique DTs**  In contrast to vanilla DTs that make hard decision at each internal node, many approaches to hierarchical mixture of expert models (Jordan & Jacobs, 1994) have been proposed. They make soft decisions where each branch is associated with a probability (Irsoy et al., 2012; Frosst & Hinton, 2017). The resulting models do not comprise axis-aligned splits, but are oblique

with respect to the axis. These adjustments in the tree architecture allow the application of further optimization algorithms, as for instance the expectation–maximization (EM) algorithm for maximum likelihood estimation (Dempster et al., 1977; Jordan & Jacobs, 1994) or gradient descent (Irsoy et al., 2012; Frosst & Hinton, 2017). Blanquero et al. (2020) try to increase the interpretability of oblique trees by optimizing for sparse splits using fewer predictor variables at each split and simultaneously fewer splits along the whole tree. Tanno et al. (2019) combine the benefits of neural networks and DTs using so-called adaptive neural trees (ANTs) that allow a gradient-based end-to-end optimization. They employ a stochastic routing based on a Bernoulli distribution where the mean is a learned parameter. Furthermore, ANTs utilize one or more non-linear transformer modules at the edges, making the resulting trees oblique. Norouzi et al. (2015) proposed an approach for efficient non-greedy optimization of DTs that overcomes the need of soft decisions to apply gradient-based algorithms. This includes a joint optimization of the split function at all levels and leaf parameters by minimizing a convex-concave upper bound on the tree's empirical loss. While this allows the use of hard splits, the approach is still limited to oblique trees. Zantedeschi et al. (2021) use argmin differentiation to simultaneously learn all tree parameters by relaxing a mixed-integer program for discrete parameters to allow a gradient-based optimization. This allows a deterministic tree routing which is similar to GDTs. However, they require a differentiable splitting function (e.g. a linear split function which results in oblique trees). In contrast to oblique DTs, GDTs make axis-aligned splits that only consider a single feature at each split, which provides a significantly higher interpretability for individual splits. This is supported by Molnar (2020) where the author argues that humans cannot comprehend explanations involving more than three dimensions at once.

**Deep Neural Decision Trees (DNDTs)**  Yang et al. (2018) propose DNDTs which realize tree models as neural networks, utilizing a soft binning function for splitting. Therefore, the resulting trees are soft[2], but axis-aligned which makes this work closely related to our approach. Since the tree is generated via Kronecker product of the binning layers the structure depends on the number of features (and the number of bins). As the authors discussed, this results in a poor scalability with the number of features, which can currently only be solved by using random forests for high-dimensional datasets ($\geq 12$ features). Our approach, in contrast, scales linearly with the number of features, making it applicable for high-dimensional datasets as well.

## 3  GRADIENT-BASED AXIS-ALIGNED DECISION TREES

In this section, we propose gradient-based decision trees (GDTs) as a novel approach to learn DTs. Therefore, we introduce a novel DT representation and a corresponding algorithm that allows learning hard, axis-aligned DTs with gradient descent. More specifically, we will use backpropagation with a straight-through (ST) operator (Section 3.3) on a dense DT representation (Section 3.2) to adjust the model parameters during the training, as we will explain in Section 3.4.

### 3.1  ARITHMETIC DECISION TREE REPRESENTATION

In the following, we will introduce a notation for DTs with respect to their parameters. We formulate DTs as an arithmetic function based on addition and multiplication instead of a nested concatenation of rules. This formulation will be useful for describing the gradient-based learning in Sections 3.2-3.4. Note that we focus on learning fully-grown DTs within this paper.

The parameters of a DT of depth $d$ comprise one split threshold and one feature index for each internal node, which we denote as vectors $\boldsymbol{\tau} \in \mathbb{R}^{2^d-1}$ and $\boldsymbol{\iota} \in \mathbb{N}^{2^d-1}$ respectively, where $2^d - 1$ equals the number of internal nodes in a fully-grown DT.Additionally, each leaf node comprises a class membership in the case of a classification task or a value in the case of regression tasks, which we denote as the vector $\boldsymbol{\lambda} \in \mathcal{C}^{2^d}$, where $\mathcal{C}$ is the set of classes and $2^d$ equals the number of leaf nodes in a fully-grown DTs.Figure 2 shows our tree representation in comparison to a common tree representation for an exemplary DT.

---

[2]The authors also propose using ST Gumbel-Softmax as alternative to generate hard trees, which is similar to our approach.

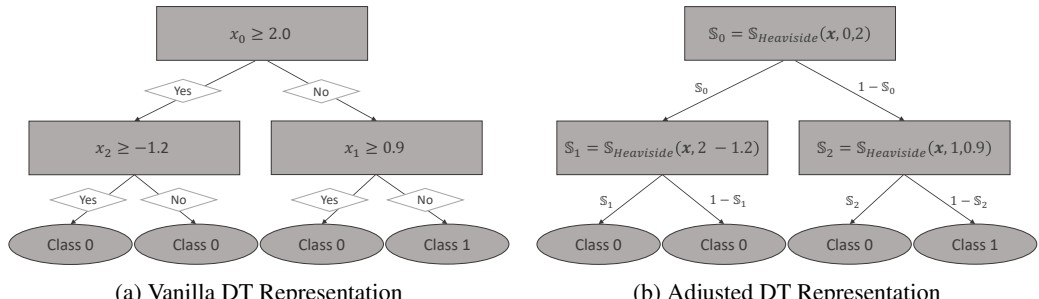

(a) Vanilla DT Representation        (b) Adjusted DT Representation

Figure 2: **DT Representations.** Exemplary DT representations with depth 2 for a dataset comprising 3 variables and 2 classes. The DT at the right can be represented using the vectors $\boldsymbol{\iota} = [0, 2, 1]$, $\boldsymbol{\tau} = [2.0, -1.2, 0.9]$ and $\boldsymbol{\lambda} = [0, 0, 0, 1]$.

Formally, we can express a DT as a function $DT(\cdot|\boldsymbol{\tau}, \boldsymbol{\iota}, \boldsymbol{\lambda}) : \mathbb{R}^n \to \mathcal{C}$ with respect to its parameters:

$$DT(\boldsymbol{x}|\boldsymbol{\tau}, \boldsymbol{\iota}, \boldsymbol{\lambda}) = \sum_{l=0}^{2^d} \lambda_l \, \mathbb{L}(\boldsymbol{x}|l, \boldsymbol{\tau}, \boldsymbol{\iota}) \tag{1}$$

$\mathbb{L}$ is a function that indicates whether a sample $\boldsymbol{x} \in \mathbb{R}^n$ belongs to a leaf $l$.

The indicator function $\mathbb{L}$ can be defined as a multiplication of the split functions of the preceding internal nodes. We define a split function $\mathbb{S}$ as a Heaviside step function defined as:

$$\mathbb{S}_{\text{Heaviside}}(\boldsymbol{x}|\iota, \tau) = \begin{cases} 1, \text{if } x_\iota \geq \tau \\ 0, \text{otherwise} \end{cases} \tag{2}$$

where $\iota$ is the index of the feature considered at a certain split and $\tau$ is the corresponding split threshold.

Enumerating the internal nodes of a fully-grown tree with depth $d$ in a breadth-first order, we can now define the indicator function $\mathbb{L}$ as:

$$\begin{aligned} \mathbb{L}(\boldsymbol{x}|l, \boldsymbol{\tau}, \boldsymbol{\iota}) = \prod_{j=1}^{d} & (1 - \mathfrak{p}(l, d, j)) \, \mathbb{S}(\boldsymbol{x}|\tau_{\mathfrak{i}(l,d,j)}, \iota_{\mathfrak{i}(l,d,j)}) \\ & + \mathfrak{p}(l, d, j) \left(1 - \mathbb{S}(\boldsymbol{x}|\tau_{\mathfrak{i}(l,d,j)}, \iota_{\mathfrak{i}(l,d,j)})\right) \end{aligned} \tag{3}$$

Here, $\mathfrak{i}$ is the index of the internal node preceding a specific leaf node $l$ at a certain depth $j$. Further, $\mathfrak{p}$ defines whether the left branch ($\mathfrak{p} = 0$) or the right branch ($\mathfrak{p} = 1$) was taken at a certain internal node to reach a leaf node $l$. Both values, $\mathfrak{i}$ and $\mathfrak{p}$, are constant for a certain architecture and can be calculated straightforward (see Appendix A.1).

As becomes evident, DTs involve non-differentiable operations (Equation 2), which precludes the application of the backpropagation algorithm for learning the parameters. More specifically, there are three challenges we need to solve if we want to use backpropagation to efficiently learn a DT:

**C1** The index $\iota$ for the selection of the considered feature in a certain split is defined as $\iota \in \mathbb{N}$. However, the index $\iota$ is a parameter of the DT and a standard optimization with gradient descent requires $\iota \in \mathbb{R}$.

**C2** The split function $\mathbb{S}(\boldsymbol{x}, \iota, \tau)$ is a Heaviside step function with an undefined gradient for $x_\iota = \tau$ and 0 gradient elsewhere, which precludes an efficient optimization.

**C3** The leaf node in a vanilla DT comprises only the class membership and therefore $\lambda \in \mathcal{C}$. To optimize the leaf node parameters, we need $\lambda \in \mathbb{R}$ to apply gradient descent and calculate an informative loss value as cost function.

The main difference of GDTs to a standard DT algorithm is that we use a dense representation for the internal node parameters ($\boldsymbol{\tau}$ and $\boldsymbol{\iota}$) to jointly optimize the split thresholds and the selection of the corresponding features for all internal nodes (Section 3.2). Furthermore, we use a ST operator during the backpropagation to allow the use of hard and axis-aligned DTs during training (Section 3.3).

## 3.2 Dense Decision Tree Representation

In this subsection, we propose a differentiable representation of the feature indices $\boldsymbol{\iota}$ to allow a gradient-based optimization. Therefore, we extend the vector $\boldsymbol{\iota} \in \mathbb{R}^{2^d-1}$ to a matrix, where $I \in \mathbb{R}^{2^d-1} \times \mathbb{R}^n$. This is achieved by one-hot encoding the feature index as $\boldsymbol{\iota} \in \mathbb{R}^n$ for each internal node. This adjustment is necessary for the optimization process to account for the fact that feature indices are categorical instead of ordinal.

We further propose using a similar representation for the split thresholds as $T \in \mathbb{R}^{2^d-1} \times \mathbb{R}^n$ by storing one value for each feature as $\boldsymbol{\tau} \in \mathbb{R}^n$ instead of a single feature. This adjustment is designed to support the optimization procedure, since the split thresholds are not interchangeable for different features: A split threshold that is reasonable for one feature might not be reasonable for another feature. By storing one split threshold for each feature, we support exploration during the optimization.

This representation is inspired by the DT representation of Marton et al. (2022) used for predicting a DT as a numeric output of a neural network. Further, our matrix representation for the feature selection is similar to the one proposed by Popov et al. (2019) but does additionally include a matrix representation for the split thresholds.

Besides the previously mentioned advantages, using a dense DT representation allows the use of matrix multiplications for an efficient computation, as we will show in the following. Therefore, we reformulate the Heaviside split function in Equation 2 as follows:

$$\mathbb{S}_{\text{logistic}}(\boldsymbol{x}, \boldsymbol{\iota}, \boldsymbol{\tau}) = \sigma \left( \sum_{i=0}^{n} \iota_i x_i - \sum_{i=0}^{n} \iota_i \tau_i \right) \tag{4}$$

$$\mathbb{S}_{\text{logistic\_hard}}(\boldsymbol{x}, \boldsymbol{\iota}, \boldsymbol{\tau}) = \lfloor \mathbb{S}_{\text{logistic}}(\boldsymbol{x}, \boldsymbol{\iota}, \boldsymbol{\tau}) \rceil \tag{5}$$

where $\lfloor \cdot \rceil$ stands for rounding to the closest integer. For our case, where $\boldsymbol{\iota}$ is a one-hot encoded vector for the feature index, $\mathbb{S}_{\text{logistic\_hard}}(\boldsymbol{x}, \boldsymbol{\iota}, \boldsymbol{\tau}) = \mathbb{S}_{\text{Heaviside}}(\boldsymbol{x}, \iota, \tau)$ holds.

## 3.3 Backpropagation of Decision Tree Loss

While the dense DT representation introduced in the previous section emphasizes an efficient learning of axis-aligned DTs, it does not solve **C1**-**C3**. Therefore, in this subsection, we will explain how we solve those challenges by using a ST operator during the backpropagation when optimizing with gradient descent.

For the function value calculation in the forward pass, we need to assure that $\boldsymbol{\iota}$ is a one-hot encoded vector. This can be achieved by applying a hardmax function on the feature index vector of each internal node. However, applying a hardmax is a non-differentiable operation, which precludes gradient computation. To overcome this issue, we use the ST operator (Bengio et al., 2013): For the forward pass, we apply the hardmax as is. For the backward pass, however, we exclude this operation and directly propagate back the gradients of $\boldsymbol{\iota}$. Accordingly, we can optimize the parameters of $\boldsymbol{\iota}$ where $\boldsymbol{\iota} \in \mathbb{R}$ while still using axis-aligned splits during training (**C1**).

Similarly, we use the ST operator to assure hard splits (Equation 5) by excluding $\lfloor \cdot \rceil$ for the backward pass (**C2**). Using the sigmoid logistic function before applying the ST operator (see Equation 4) utilizes the distance to the split threshold as additional information for the gradient calculation. If the feature considered at an internal node for a specific sample is close to the split threshold, this will result in smaller gradients compared to a sample that is more distant to the split threshold.

Furthermore, we need to adjust the leaf nodes of the DT to allow an efficient loss calculation (**C3**). Vanilla DTs comprise the predicted class for each leaf node and are functions $DT : \mathbb{R}^n \to \mathcal{C}$. We use

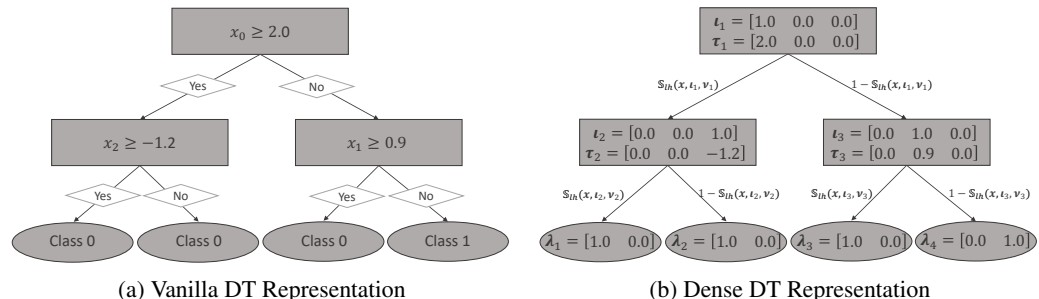

(a) Vanilla DT Representation         (b) Dense DT Representation

Figure 3: **DT Representations.** Exemplary DT representations with depth 2 for a dataset comprising 3 variables and 2 classes. Here, $\mathbb{S}_{lh}$ stands for $\mathbb{S}_{\text{logistic\_hard}}$ (Equation 5).

a probability distribution at each leaf node and therefore define DTs as a function $DT : \mathbb{R}^n \to \mathbb{R}^c$. Accordingly, the parameters of the leaf nodes are defined as $L \in \mathbb{R}^{2^n} \times \mathbb{R}^c$ for the whole tree and $\boldsymbol{\lambda} \in \mathbb{R}^c$ for a specific leaf node. This adjustment allows the application of standard loss functions, as for instance the cross-entropy, during the optimization.

Figure 3 visualizes our dense tree representation in comparison to standard tree representation for an exemplary DT.

### 3.4 Training Procedure

In the previous subsections, we introduced the adjustments that are necessary to apply a gradient-based optimization for DTs. We implement this optimization using a gradient descent algorithm. During the gradient descent optimization, we calculate the gradients using backpropagation (see Algorithm 2). We implement backpropagation using automatic differentiation based on the computation graph of the tree pass function, which is used to calculate the function values. The tree pass function is summarized in Algorithm 1 and utilizes the adjustments introduced in the previous sections. Furthermore, our tree routing allows calculating the tree pass function as a joint matrix operation for all tree nodes and all samples. We also want to note that the sophisticated dense representation which is necessary during the training can be converted into an equivalent vanilla DT representation at each point in time.

Furthermore, our implementation optimizes the gradient descent algorithm by exploiting common stochastic gradient descent techniques, including mini-batch calculation and momentum using the Adam optimizer (Kingma & Ba, 2014). Moreover, we implement an early stopping procedure and to avoid bad initial parametrizations during the initialization, we additionally implement random restarts where the best model is selected based on the validation loss.

## 4 Experimental Evaluation

The goal of our experiments is to evaluate the predictive performance of GDTs against existing approaches to learning DTs. We focus on axis-aligned DTs and compare GDTs to the following baseline algorithms:

- **CART**: We use the sklearn (Pedregosa et al., 2011) implementation, which uses an optimized version of the CART algorithm. While CART usually only uses Gini as impurity measure, we also allow information gain/entropy as an option during our hyperparameter optimization (HPO). Since the impurity measure is also the relevant difference between CART and C4.5, we decided to not use both algorithms as a benchmark, but only the optimized CART algorithm which is also the stronger benchmark to compare against.

- **Evolutionary DTs**: We use the GeneticTree (Pysiak, 2021) implementation for evolutionary DTs that implements an efficient learning of DTs using a genetic algorithm.

- **DNDT** (Yang et al., 2018): We use the official impmementation (Yang et al., 2022) for learning DTs with gradient descent.

---

**Algorithm 1** Tree Pass of a Training Sample

---

1: **function** PASS($I, T, L, \boldsymbol{x}$)
2: $\quad c_1^* \leftarrow I_i - \text{hardmax}(I_i) \quad \text{for } i = 0, \ldots, |I| \qquad\qquad$ ▷ Excluded in the backward pass
3: $\quad I \leftarrow I - c_1^* \qquad\qquad$ ▷ Excluded in the backward pass
4: $\quad \hat{\boldsymbol{y}} \leftarrow [0]^c$
5: $\quad$ **for** $l = 0, \ldots, 2^d$ **do**
6: $\qquad p \leftarrow 1$
7: $\qquad$ **for** $j = 1, \ldots, d$ **do**
8: $\qquad\quad \mathfrak{i} \leftarrow 2^{j-1} + \left\lfloor \frac{l}{2^{d-(j-1)}} \right\rfloor - 1 \qquad\qquad$ ▷ Equation 6
9: $\qquad\quad \mathfrak{p} \leftarrow \left\lfloor \frac{l}{2^{d-j}} \right\rfloor \bmod 2 \qquad\qquad$ ▷ Equation 7
10: $\qquad\quad s \leftarrow \sigma\left(\sum_{i=0}^{n} T_{\mathfrak{i},i}\, I_{\mathfrak{i},i} - \sum_{i=0}^{n} x_i\, I_{\mathfrak{i},i}\right) \qquad\qquad$ ▷ Equation 4
11: $\qquad\quad c_2^* = s - \lfloor s \rceil \qquad\qquad$ ▷ Excluded in the backward pass
12: $\qquad\quad s \leftarrow s - c_2^* \qquad\qquad$ ▷ Excluded in the backward pass
13: $\qquad\quad p \leftarrow p\left((1-\mathfrak{p})\, s + \mathfrak{p}\,(1-s)\right) \qquad\qquad$ ▷ Equation 3
14: $\qquad$ **end for**
15: $\qquad \hat{\boldsymbol{y}} \leftarrow \hat{\boldsymbol{y}} + L_l\, p \qquad\qquad$ ▷ Equation 1
16: $\quad$ **end for**
17: $\quad$ **return** $\hat{\boldsymbol{y}}$
18: **end function**

---

- **DL8.5** (Aglin et al., 2020): We use the official implementation (Aglin et al., 2022) for learning optimal DTs which includes improvements from MurTree (Demirović et al., 2022) reducing the runtime significantly. As required, we discretized the data for DL8.5 as described in the Appendix A.4.

GDTs are implemented in Python using TensorFlow (Abadi et al., 2015)[3]. The experiments were conducted on several benchmark datasets, mainly from the UCI Machine Learning repository (Dua & Graff, 2017). The dataset specifications and source are listed in Table 7. We use a random $80\%/20\%$ split to train and test data for all datasets. Since GDTs and DNDTs requires a validation set for early stopping, we performed another $80\%/20\%$ split on the training data. The remainder of the approaches utilize the complete training data. Further information on the hyperparameters is given in the Appendix A.5.

## 4.1 RESULTS

**GDTs are competitive to baseline DT learners**    First, we evaluated the performance of GDTs against the baseline approaches on the mentioned benchmark datasets in terms of the F1-Score. We noted the mean reciprocal rank (MRR), similar to Yang et al. (2018). The results are shown in Table 1. Overall, GDTs outperformed state-of-the-art non-greedy DT approaches for binary classification tasks (MRR of $0.648$ for GDT vs. $0.454$ for GeneticTree as best non-greedy benchmark) and achieved competitive results for multi-class tasks ($0.512$ for GDT vs. $0.490$ for DNDT). Compared to a greedy DT trained by improved CART, GDTs achieved a slightly higher performance on binary classification tasks ($0.648$ vs. $0.623$) but underperformed on multi-class tasks ($0.512$ vs. $0.685$). Considering the stdev of the reciprocal rank, we can observe that GDTs ($0.275$ and $0.213$) are more robust than to CART ($0.323$ and $0.282$).

Further, we can observe that for several datasets a greedy optimization achieved very poor results and was significantly outperformed by a non-greedy optimization, as for instance on the *Wisconsin Diagnostic Breast Cancer*, *Heart Disease* and *Lymphography* datasets.

In general, it stands out, that GDTs achieved the best results for binary classification datasets with a lower dimensionality ($n \leq 15$). We can explain this by the dense DT representation required for the gradient-based optimization. Using our representation, the difficulty of the optimization task increases with the number of features (more parameters at each internal node) and the number of

---

[3]The code of our implementation is available to all reviewers in the supplementary material to assure anonymity. We will make it publically available here upon acceptance.

| | GDT | Greedy (CART) | GeneticTree | DNDT | DL8.5 |
|---|---|---|---|---|---|
| Blood Transfusion | **0.763 ± 0.029 (1)** | 0.743 ± 0.042 (2) | 0.684 ± 0.049 (5) | 0.726 ± 0.046 (3) | 0.725 ± 0.030 (4) |
| Banknote Authentication | **0.988 ± 0.008 (1)** | 0.982 ± 0.006 (2) | 0.928 ± 0.024 (5) | 0.945 ± 0.015 (4) | 0.963 ± 0.011 (3) |
| Titanic | 0.808 ± 0.030 (2) | 0.802 ± 0.033 (3) | 0.779 ± 0.033 (4) | 0.736 ± 0.069 (5) | **0.816 ± 0.027 (1)** |
| Raisins | 0.852 ± 0.015 (3) | 0.853 ± 0.017 (2) | **0.857 ± 0.021 (1)** | 0.852 ± 0.032 (4) | 0.849 ± 0.027 (5) |
| Rice | 0.928 ± 0.006 (2) | 0.928 ± 0.004 (3) | 0.928 ± 0.005 (4) | **0.929 ± 0.006 (1)** | 0.926 ± 0.008 (5) |
| Echocardiogram | 0.722 ± 0.109 (4) | **0.730 ± 0.086 (1)** | 0.723 ± 0.079 (3) | 0.728 ± 0.073 (2) | 0.679 ± 0.100 (5) |
| Wisconsin Diagnostic Breast Cancer | **0.920 ± 0.029 (1)** | 0.894 ± 0.025 (5) | 0.901 ± 0.024 (4) | 0.918 ± 0.028 (2) | 0.903 ± 0.021 (3) |
| Loan House | **0.769 ± 0.030 (1)** | 0.767 ± 0.036 (3) | 0.769 ± 0.033 (2) | 0.665 ± 0.052 (5) | 0.757 ± 0.026 (4) |
| Heart Failure | **0.791 ± 0.040 (1)** | 0.778 ± 0.057 (3) | 0.779 ± 0.062 (2) | 0.627 ± 0.074 (5) | 0.720 ± 0.060 (4) |
| Heart Disease | 0.798 ± 0.042 (2) | 0.755 ± 0.064 (4) | **0.803 ± 0.043 (1)** | - | 0.771 ± 0.050 (3) |
| Adult | 0.839 ± 0.003 (2) | **0.848 ± 0.004 (1)** | 0.794 ± 0.003 (4) | - | 0.819 ± 0.003 (3) |
| Bank Marketing | 0.859 ± 0.011 (3) | **0.868 ± 0.004 (1)** | 0.827 ± 0.004 (4) | - | 0.864 ± 0.004 (2) |
| Cervical Cancer | **0.920 ± 0.019 (1)** | 0.917 ± 0.022 (3) | **0.920 ± 0.019 (1)** | - | 0.910 ± 0.022 (4) |
| Congressional Voting | 0.949 ± 0.021 (3) | **0.960 ± 0.017 (1)** | **0.960 ± 0.017 (1)** | - | 0.941 ± 0.025 (4) |
| Absenteeism | **0.691 ± 0.039 (1)** | 0.688 ± 0.031 (2) | 0.662 ± 0.043 (4) | - | 0.676 ± 0.035 (3) |
| Hepatitis | 0.785 ± 0.112 (2) | 0.782 ± 0.071 (4) | 0.782 ± 0.071 (3) | - | **0.795 ± 0.076 (1)** |
| German | 0.690 ± 0.038 (2) | **0.699 ± 0.026 (1)** | 0.636 ± 0.038 (4) | - | 0.687 ± 0.028 (3) |
| Mushroom | **1.000 ± 0.000 (1)** | **1.000 ± 0.000 (1)** | 0.967 ± 0.013 (4) | - | **1.000 ± 0.000 (1)** |
| Credit Card | 0.798 ± 0.006 (2) | **0.799 ± 0.005 (1)** | 0.795 ± 0.005 (4) | - | 0.796 ± 0.007 (3) |
| Horse Colic | 0.840 ± 0.047 (2) | 0.840 ± 0.034 (3) | **0.868 ± 0.027 (1)** | - | 0.831 ± 0.035 (4) |
| Thyroid | 0.937 ± 0.007 (2) | **0.963 ± 0.003 (1)** | 0.831 ± 0.031 (3) | - | 0.797 ± 0.008 (4) |
| Spambase | 0.892 ± 0.012 (2) | **0.923 ± 0.011 (1)** | 0.874 ± 0.017 (3) | - | 0.869 ± 0.012 (4) |
| Mean Reciprocal Rank | **0.648 ± 0.275 (1)** | 0.623 ± 0.323 (2) | 0.454 ± 0.306 (3) | 0.381 ± 0.246 (5) | 0.383 ± 0.254 (4) |
| Iris | **0.944 ± 0.047 (1)** | 0.930 ± 0.032 (3) | 0.890 ± 0.061 (5) | 0.933 ± 0.034 (2) | 0.913 ± 0.043 (4) |
| Balance Scale | 0.737 ± 0.029 (3) | 0.773 ± 0.022 (2) | 0.713 ± 0.049 (4) | **0.776 ± 0.024 (1)** | 0.661 ± 0.045 (5) |
| Car | 0.852 ± 0.038 (2) | **0.975 ± 0.003 (1)** | 0.715 ± 0.044 (4) | 0.701 ± 0.042 (5) | 0.827 ± 0.024 (3) |
| Glass | 0.678 ± 0.074 (2) | **0.681 ± 0.076 (1)** | 0.587 ± 0.102 (5) | 0.671 ± 0.082 (3) | 0.634 ± 0.069 (4) |
| Contraceptive | 0.531 ± 0.026 (3) | 0.559 ± 0.025 (2) | 0.503 ± 0.048 (4) | 0.477 ± 0.048 (5) | **0.568 ± 0.020 (1)** |
| Solar Flare | 0.793 ± 0.027 (3) | 0.795 ± 0.024 (2) | 0.793 ± 0.026 (4) | **0.801 ± 0.028 (1)** | 0.790 ± 0.026 (5) |
| Wine | **0.923 ± 0.041 (1)** | 0.914 ± 0.032 (2) | 0.891 ± 0.055 (3) | 0.847 ± 0.046 (5) | 0.856 ± 0.018 (4) |
| Zoo | 0.946 ± 0.054 (2) | **0.970 ± 0.050 (1)** | 0.850 ± 0.100 (4) | - | 0.929 ± 0.087 (3) |
| Lymphography | 0.769 ± 0.050 (2) | 0.724 ± 0.075 (4) | 0.740 ± 0.071 (3) | - | **0.824 ± 0.102 (1)** |
| Segment | 0.943 ± 0.009 (2) | **0.966 ± 0.005 (1)** | 0.770 ± 0.049 (4) | - | 0.810 ± 0.013 (3) |
| Dermatology | 0.934 ± 0.032 (2) | **0.965 ± 0.023 (1)** | 0.894 ± 0.102 (4) | - | 0.929 ± 0.026 (3) |
| Landsat | 0.852 ± 0.008 (2) | **0.856 ± 0.008 (1)** | 0.674 ± 0.041 (4) | - | 0.808 ± 0.006 (3) |
| Annealing | 0.970 ± 0.014 (3) | 0.973 ± 0.011 (2) | 0.821 ± 0.091 (4) | - | **0.985 ± 0.007 (1)** |
| Splice | 0.883 ± 0.011 (3) | 0.918 ± 0.006 (2) | 0.667 ± 0.066 (4) | - | **0.923 ± 0.015 (1)** |
| Mean Reciprocal Rank | 0.512 ± 0.213 (2) | **0.685 ± 0.282 (1)** | 0.255 ± 0.036 (5) | 0.490 ± 0.337 (3) | 0.487 ± 0.328 (4) |

Table 1: **Performance Comparison.** We report F1-scores (mean ± stdev over 10 trials). We also report the ranking of each approach in brackets. The top part comprises binary classification tasks and the bottom part multi-class datasets. The datasets are sorted by the number of features. Due to scalability issues, DNDTs were only applied to datasets with a maximum of 12 features, as suggested by Yang et al. (2018).

classes (more parameters at each leaf node). Therefore, in future work, we aim to optimize the proposed dense representation, e.g., by using parameter sharing techniques.

**GDTs are robust to overfitting** We can observe that GDTs were more robust and less prone to overfitting than a greedy optimization. We measure overfitting by the difference between the mean train and test performance (see Table 2). This difference was significantly smaller for GDTs (0.026 for binary and 0.038 for multi-class) compared to CART (0.042 and 0.085). An even higher overfitting can be observed for DNDT (0.081 and 0.062) and DL8.5 (0.069 and 0.059). Overfitting can also explain why optimal decision trees, do not achieve superior results on test data, which was also reported by Zharmagambetov et al. (2021). Similar to GDTs, GeneticTrees were less prone to overfitting (0.015 and 0.021).

**GDTs do not rely on extensive HPO** An advantage of DTs over more sophisticated models, besides their interpretability, is that greedy approaches are not reliant on an extensive HPO to achieve reasonable results. In this experiment, we wanted to show that the same is true for GDTs. We used the default parameter settings for each model and fixed the maximum depth to 5 for GDTs and CART. Overall, the performance gain of HPO was rather small for all approaches: The maximum average performance gain for binary tasks could be observed for CART, with an increase of only 0.007 (see Table 2). For multi-class tasks, the average performance gain was higher with a maximum of 0.049 for GDTs. However, this is mainly due to a small number of datasets where HPO increased the performance significantly as for instance the *Segment* dataset with a performance increase of 0.111 for GDTs and 0.171 for CART (see Table 3 for details).

| Model | Train-Test Differential (Mean) | | HPO Impact (Mean) | |
|-------|:-----:|:-----:|:-----:|:-----:|
| | binary | multi-class | binary | multi-class |
| GDT | 0.026 | 0.038 | 0.006 | 0.049 |
| Greedy (CART) | 0.042 | 0.085 | 0.007 | 0.032 |
| GeneticTree | 0.015 | 0.021 | 0.000 | 0.004 |
| DNDT | 0.081 | 0.062 | 0.000 | 0.000 |
| DL8.5 | 0.069 | 0.054 | 0.000 | 0.000 |

Table 2: **Summarized Results.** (1) The mean difference between the train and test performance of each approach as overfitting indicator (see Table 6 for complete results) and (2) the impact of HPO by measuring the difference of the model performance with and without HPO for each approach (see Table 3 for complete results).

**Probabilities at the GDTs leaf nodes provide calibrated uncertainties** Another advantage of GDTs (and similarly DNDTs) is the fact that they provide probabilities as a direct measure of the model's confidence. We compared the probability-based confidence with the purity measure[4] in terms of the ROC AUC score (see Table 5). Comparing the probability-based approaches, GDTs achieved a significantly higher MRR based on the ROC AUC score of $0.818$ for binary and $0.789$ for multi-class tasks, compared to DNDT ($0.628$ and $0.655$). Both of these probability-based approaches significantly outperformed approaches that learn vanilla DTs, where CART achieved the highest MRR ($0.460$ and $0.458$).

**GTDs can be learned efficiently for large and high-dimensional datasets** Considering the runtime of the different approaches listed in Table 4, it stands out, that a greedy optimization was significantly faster than any other approach for each dataset (perfect MRR of $1.00$ based on the runtime). Nevertheless, for most datasets, learning a GDT took less than 60 seconds, with a maximum of 122 seconds for the *Congressional Voting* dataset. DNDTs achieved similar runtimes compared to GDTs. This is also reflected by the MRR where GDTs achieved $0.254$ for binary and $0.257$ for multi-class and DNDTs achieved $0.228$ and $0.221$ respectively. DL8.5 has a low runtime $< 10$ seconds for most datasets, however we can observe scalability issues with the number of features and number of samples as they require a significantly higher runtime for certain datasets (e.g., $> 330$ seconds for *Credit Card* and $> 1800$ seconds for *Splice*).

## 5 CONCLUSION AND FUTURE WORK

In this paper, we proposed a method for learning hard, axis-aligned DTs based on a joint optimization of all parameters with gradient descent. We use backpropagation with a ST operator to deal with the non-differentiable nature of DTs and introduced a dense DT representation that allows an efficient optimization. Using a gradient-based optimization, GDTs are not prone to locally optimal solutions, as it is the case for standard, greedy DT induction methods like CART. We empirically showed that GDTs outperform existing baseline methods on several benchmark datasets. Additionally, GDTs provide calibrated uncertainties in terms of probability distributions at the leaf nodes, which also increases the interpretability of the model.

Furthermore, a gradient-based optimization provides more flexibility. It is straightforward to use a custom loss function that is well-suited to the specific application scenario. Another advantage of a gradient-based optimization is the possibility to relearn the threshold value as well as the split index. This allows the application of GDTs to dynamic environments, as for instance online learning scenarios, without adjustments.

Currently, GDTs are fully-grown. In future work, we want to apply pruning mechanisms to reduce the tree size for instance through a learnable choice parameter to decide if a node is pruned, similar to Zantedeschi et al. (2021). While we focused on stand-alone DTs to generate intrinsically interpretable models within this paper, GDTs can easily be extended to random forests as a performance-interpretability trade-off, which is subject to future work.

---

[4]Commonly, probabilities are obtained from the purity ratio of each class as a gateway to the model's confidence for vanilla DTs (Pedregosa et al., 2011).

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

# A APPENDIX

## A.1 CALCULATIONS

The index $\mathfrak{i}$ of an internal node preceding a specific leaf node $l$ at a certain depth $j$ can be calculated as follows:

$$\mathfrak{i}(l, d, j) = 2^{j-1} + \left\lfloor \frac{l}{2^{d-(j-1)}} \right\rfloor - 1 \tag{6}$$

Additionally, for a certain leaf node $l$, $\mathfrak{p}$ defines whether the left branch ($\mathfrak{p} = 0$) or the right branch ($\mathfrak{p} = 1$) was taken at the internal node $\mathfrak{i}$. We can calculate $\mathfrak{p}$ as follows:

$$\mathfrak{p}(l, d, j) = \left\lfloor \frac{l}{2^{d-j}} \right\rfloor \mod 2 \tag{7}$$

The calculation of the internal node index $\mathfrak{i}$ as well as the specification of the path position $\mathfrak{p}$ also involve non-differentiable operations. However, since we only focus on fully-grown trees, the resulting values are constant and can be calculated independently of the optimization, which does not preclude the application of a gradient-based optimization algorithm.

## A.2 GRADIENT DESCENT OPTIMIZATION

We use stochastic gradient descent (SGD) to minimize the loss function of GDTs, which is outlined in Algorithm 2. We use backpropagation to calculate the gradients in Line 11-13. Furthermore, our implementation optimizes Algorithm 2 by exploiting common SGD techniques, including mini-batch calculation and momentum using the Adam optimizer Kingma & Ba (2014). We also formulate Line 6-9 as a single matrix multiplication for efficiency reasons.

---

**Algorithm 2** Gradient Descent Training for Decision Trees

1: **function** TRAINDT$(I, T, L, X, \boldsymbol{y}, n, c, d, \xi)$
2: $\quad I \sim \mathcal{U}\left(-\sqrt{\frac{6}{2^{2d-1}+n}}, \sqrt{\frac{6}{2^{2d-1}+n}}\right)$         ▷ Glorot uniform initialization
3: $\quad T \sim \mathcal{U}\left(-\sqrt{\frac{6}{2^{2d-1}+n}}, \sqrt{\frac{6}{2^{2d-1}+n}}\right)$         ▷ Glorot uniform initialization
4: $\quad L \sim \mathcal{U}\left(-\sqrt{\frac{6}{2^{2d}+c}}, \sqrt{\frac{6}{2^{2d}+c}}\right)$         ▷ Glorot uniform initialization
5: $\quad$ **for** $i = 1, \ldots, \xi$ **do**
6: $\quad\quad \hat{\boldsymbol{y}} \leftarrow \emptyset$
7: $\quad\quad$ **for** $j = 1, \ldots, |X|$ **do**
8: $\quad\quad\quad \hat{y}_j = \text{PASS}(I, T, L, X_j)$
9: $\quad\quad\quad \hat{\boldsymbol{y}} \leftarrow \hat{\boldsymbol{y}} \cup \hat{y}_j$
10: $\quad\quad$ **end for**
11: $\quad\quad I \leftarrow I + \eta \frac{\partial}{\partial I} \mathcal{L}(\boldsymbol{y}, \hat{\boldsymbol{y}})$         ▷ Calculate gradients with backpropagation
12: $\quad\quad T \leftarrow T + \eta \frac{\partial}{\partial T} \mathcal{L}(\boldsymbol{y}, \hat{\boldsymbol{y}})$         ▷ Calculate gradients with backpropagation
13: $\quad\quad L \leftarrow L + \eta \frac{\partial}{\partial L} \mathcal{L}(\boldsymbol{y}, \hat{\boldsymbol{y}})$         ▷ Calculate gradients with backpropagation
14: $\quad$ **end for**
15: **end function**

---

## A.3 ADDITIONAL RESULTS

## A.4 DATASETS

The datasets along with their specifications and source are summarized in Table 7. For all datasets, we performed a standard preprocessing. We applied ordinal encoding to all non-numeric features so that they can be handled by DTs. We further standardized all datasets to zero mean and unit variance. This step however is only necessary for our approach and DNDTs. While standard DT

|  | GDT | Greedy (CART) | GeneticTree | DNDT | DL85 |
|---|---|---|---|---|---|
| Blood Transfusion | **0.762 ± 0.035 (1)** | 0.733 ± 0.045 (2) | 0.671 ± 0.036 (5) | 0.726 ± 0.046 (3) | 0.725 ± 0.030 (4) |
| Banknote Authentication | **0.973 ± 0.006 (1)** | 0.971 ± 0.010 (2) | 0.929 ± 0.021 (5) | 0.945 ± 0.015 (4) | 0.963 ± 0.011 (3) |
| Titanic | 0.801 ± 0.030 (2) | 0.801 ± 0.030 (3) | 0.777 ± 0.035 (4) | 0.736 ± 0.069 (5) | **0.816 ± 0.027 (1)** |
| Raisins | 0.847 ± 0.019 (4) | 0.846 ± 0.021 (5) | **0.859 ± 0.022 (1)** | 0.852 ± 0.032 (2) | 0.849 ± 0.027 (3) |
| Rice | 0.929 ± 0.007 (2) | 0.924 ± 0.008 (5) | 0.928 ± 0.005 (3) | **0.929 ± 0.006 (1)** | 0.926 ± 0.008 (4) |
| Echocardiogram | **0.732 ± 0.093 (1)** | 0.717 ± 0.093 (4) | 0.724 ± 0.071 (3) | 0.728 ± 0.073 (2) | 0.679 ± 0.100 (5) |
| Wisconsin Diagnostic Breast Cancer | 0.911 ± 0.017 (3) | 0.917 ± 0.029 (2) | 0.897 ± 0.021 (5) | **0.918 ± 0.028 (1)** | 0.903 ± 0.021 (4) |
| Loan House | 0.771 ± 0.031 (2) | 0.747 ± 0.042 (4) | **0.775 ± 0.031 (1)** | 0.665 ± 0.052 (5) | 0.757 ± 0.026 (3) |
| Heart Failure | 0.774 ± 0.063 (2) | **0.776 ± 0.053 (1)** | 0.767 ± 0.049 (3) | 0.627 ± 0.074 (5) | 0.720 ± 0.060 (4) |
| Heart Disease | 0.780 ± 0.064 (2) | 0.726 ± 0.051 (4) | **0.820 ± 0.049 (1)** | - | 0.771 ± 0.050 (3) |
| Adult | 0.823 ± 0.015 (2) | **0.842 ± 0.005 (1)** | 0.793 ± 0.004 (4) | - | 0.819 ± 0.003 (3) |
| Bank Marketing | 0.861 ± 0.004 (3) | 0.827 ± 0.004 (2) | - | - | **0.864 ± 0.004 (1)** |
| Cervical Cancer | 0.917 ± 0.018 (2) | 0.916 ± 0.021 (3) | **0.920 ± 0.019 (1)** | - | 0.910 ± 0.022 (4) |
| Congressional Voting | 0.939 ± 0.029 (4) | 0.948 ± 0.019 (2) | **0.960 ± 0.017 (1)** | - | 0.941 ± 0.025 (3) |
| Absenteeism | 0.673 ± 0.036 (4) | **0.696 ± 0.027 (1)** | 0.678 ± 0.042 (2) | - | 0.676 ± 0.035 (3) |
| Hepatitis | **0.805 ± 0.074 (1)** | 0.777 ± 0.089 (3) | 0.770 ± 0.097 (4) | - | 0.795 ± 0.076 (2) |
| German | 0.680 ± 0.036 (3) | **0.702 ± 0.027 (1)** | 0.636 ± 0.031 (4) | - | 0.687 ± 0.028 (2) |
| Mushroom | 0.991 ± 0.008 (2) | 0.979 ± 0.002 (3) | 0.965 ± 0.010 (4) | - | **1.000 ± 0.000 (1)** |
| Credit Card | 0.795 ± 0.004 (3) | **0.798 ± 0.005 (1)** | 0.795 ± 0.005 (4) | - | 0.796 ± 0.007 (2) |
| Horse Colic | 0.835 ± 0.039 (3) | 0.849 ± 0.055 (2) | **0.862 ± 0.021 (1)** | - | 0.831 ± 0.035 (4) |
| Thyroid | 0.914 ± 0.010 (2) | **0.937 ± 0.005 (1)** | 0.840 ± 0.040 (3) | - | 0.797 ± 0.008 (4) |
| Spambase | 0.887 ± 0.015 (2) | **0.910 ± 0.009 (1)** | 0.870 ± 0.014 (3) | - | 0.869 ± 0.012 (4) |
| Mean Reciprocal Rank | 0.519 ± 0.245 (2) | **0.567 ± 0.311 (1)** | 0.478 ± 0.326 (3) | 0.465 ± 0.307 (4) | 0.411 ± 0.249 (5) |
| Iris | 0.927 ± 0.036 (3) | **0.937 ± 0.035 (1)** | 0.914 ± 0.039 (4) | 0.933 ± 0.034 (2) | 0.913 ± 0.043 (5) |
| Balance Scale | 0.722 ± 0.051 (3) | 0.736 ± 0.016 (2) | 0.700 ± 0.033 (4) | **0.776 ± 0.024 (1)** | 0.661 ± 0.045 (5) |
| Car | 0.781 ± 0.019 (3) | **0.848 ± 0.011 (1)** | 0.729 ± 0.016 (4) | 0.701 ± 0.042 (5) | 0.827 ± 0.024 (2) |
| Glass | 0.512 ± 0.126 (5) | **0.678 ± 0.074 (1)** | 0.574 ± 0.062 (4) | 0.671 ± 0.082 (2) | 0.634 ± 0.069 (3) |
| Contraceptive | 0.533 ± 0.021 (3) | 0.553 ± 0.022 (2) | 0.488 ± 0.056 (4) | 0.477 ± 0.048 (5) | **0.568 ± 0.020 (1)** |
| Solar Flare | 0.793 ± 0.027 (3) | 0.794 ± 0.031 (2) | 0.793 ± 0.026 (4) | **0.801 ± 0.028 (1)** | 0.790 ± 0.026 (5) |
| Wine | 0.902 ± 0.035 (2) | **0.914 ± 0.032 (1)** | 0.889 ± 0.045 (3) | 0.847 ± 0.046 (5) | 0.856 ± 0.018 (4) |
| Zoo | 0.867 ± 0.082 (3) | 0.921 ± 0.090 (2) | 0.828 ± 0.105 (4) | - | **0.929 ± 0.087 (1)** |
| Lymphography | 0.720 ± 0.090 (4) | 0.746 ± 0.083 (2) | 0.732 ± 0.095 (3) | - | **0.824 ± 0.102 (1)** |
| Segment | **0.832 ± 0.029 (1)** | 0.795 ± 0.024 (3) | 0.731 ± 0.074 (4) | - | 0.810 ± 0.013 (2) |
| Dermatology | 0.920 ± 0.052 (3) | **0.949 ± 0.023 (1)** | 0.865 ± 0.115 (4) | - | 0.929 ± 0.026 (2) |
| Landsat | 0.807 ± 0.020 (3) | **0.811 ± 0.009 (1)** | 0.707 ± 0.040 (4) | - | 0.808 ± 0.006 (2) |
| Annealing | 0.931 ± 0.018 (3) | 0.973 ± 0.011 (2) | 0.803 ± 0.078 (4) | - | **0.985 ± 0.007 (1)** |
| Splice | 0.821 ± 0.022 (3) | 0.900 ± 0.008 (2) | 0.703 ± 0.084 (4) | - | **0.923 ± 0.015 (1)** |
| Mean Reciprocal Rank | 0.377 ± 0.183 (4) | **0.702 ± 0.261 (1)** | 0.262 ± 0.029 (5) | 0.514 ± 0.331 (3) | 0.585 ± 0.328 (2) |

Table 3: **Performance Comparison Default Hyperparameters.** We report the F1-score (mean ± stdev over 10 trials) on the test data. We also report the ranking of each approach for the corresponding dataset in brackets. The top part comprises binary classification tasks and the bottom part multi-class datasets. The datasets are sorted based on the number of features. Due to scalability issues, DNDTs were only applied to datasets with a maximum of 12 features, as suggested by Yang et al. (2018).

learning algorithms are independent of the feature scale, GDTs and DNDTs are sensitive to the feature scale due to the gradient updates.

For DL8.5 additional preprocessing was necessary since they can only handle binary features. Therefore, we one-hot encoded all nominal and ordinal features and discretized numeric features by one-hot encoding them using quantile binning with 5 bins.

## A.5 HYPERPARAMETERS

In the following, we report the hyperparameters used for each approach. The hyperparameters were selected based on a random search over a predefined parameter range for GDT, CART and GeneticTree and are summarized in Table 8-Table 10. All parameters that were considered are noted in the tables. The number of trials was equal for each approach. For GDTs, we did not optimize the batch size as well as the number of epochs, but used early stopping with a predefined patience. For DL8.5 there are no tunable hyperparameters except the maximum depth. However, the maximum depth strongly impacts the runtime which is why we fixed the maximum depth to 4, similar to the maximum depth used during the experiments of the authors (Demirović et al., 2022; Aglin et al., 2020). Running the experiments with a higher depth becomes infeasible for many datasets. In preliminary experiments, we also observed that increasing the depth to 5 results in a decrease in the test performance. For DNDT, the number of cut points is the tunable hyperparameter of the model according to the authors (Yang et al., 2018). However, it has to be restricted to 1 in order to generate binary trees for comparability reasons. The learning rate and temperature were

| | GDT | Greedy (CART) | GeneticTree | DNDT | DL8.5 |
|---|---|---|---|---|---|
| Blood Transfusion | 27.706 ± 9.000 (5) | **0.002 ± 0.000 (1)** | 4.243 ± 1.000 (3) | 8.235 ± 4.000 (4) | 0.027 ± 0.000 (2) |
| Banknote Authentication | 46.912 ± 8.000 (4) | **0.003 ± 0.000 (1)** | 1.152 ± 0.000 (3) | 47.474 ± 37.000 (5) | 0.028 ± 0.000 (2) |
| Titanic | 20.803 ± 10.000 (5) | **0.002 ± 0.000 (1)** | 8.056 ± 5.000 (3) | 16.085 ± 4.000 (4) | 0.104 ± 0.000 (2) |
| Raisins | 42.109 ± 9.000 (5) | **0.003 ± 0.000 (1)** | 0.833 ± 0.000 (3) | 22.533 ± 7.000 (4) | 0.231 ± 0.000 (2) |
| Rice | 36.642 ± 9.000 (4) | **0.008 ± 0.000 (1)** | 1.539 ± 0.000 (3) | 71.666 ± 28.000 (5) | 0.498 ± 0.000 (2) |
| Echocardiogram | 51.105 ± 10.000 (5) | **0.002 ± 0.000 (1)** | 0.991 ± 0.000 (3) | 14.556 ± 5.000 (4) | 0.095 ± 0.000 (2) |
| Wisconsin Diagnostic Breast Cancer | 24.790 ± 7.000 (4) | **0.004 ± 0.000 (1)** | 1.987 ± 1.000 (3) | 29.417 ± 12.000 (5) | 0.554 ± 0.000 (2) |
| Loan House | 76.309 ± 26.000 (5) | **0.002 ± 0.000 (1)** | 2.191 ± 1.000 (3) | 22.476 ± 7.000 (4) | 0.194 ± 0.000 (2) |
| Heart Failure | 18.504 ± 4.000 (4) | **0.002 ± 0.000 (1)** | 0.132 ± 0.000 (2) | 35.570 ± 22.000 (5) | 0.293 ± 0.000 (3) |
| Heart Disease | 9.684 ± 15.000 (3) | **0.002 ± 0.000 (1)** | 12.946 ± 11.000 (4) | - | 0.451 ± 0.000 (2) |
| Adult | 106.875 ± 26.000 (4) | **0.086 ± 0.000 (1)** | 42.549 ± 12.000 (3) | - | 19.426 ± 7.000 (2) |
| Bank Marketing | 64.027 ± 23.000 (4) | **0.083 ± 0.000 (1)** | 6.364 ± 1.000 (2) | - | 9.339 ± 1.000 (3) |
| Cervical Cancer | 8.090 ± 2.000 (4) | **0.002 ± 0.000 (1)** | 0.211 ± 0.000 (3) | - | 0.182 ± 0.000 (2) |
| Congressional Voting | 122.195 ± 44.000 (4) | **0.002 ± 0.000 (1)** | 0.115 ± 0.000 (2) | - | 0.432 ± 0.000 (3) |
| Absenteeism | 41.944 ± 9.000 (4) | **0.003 ± 0.000 (1)** | 7.816 ± 3.000 (3) | - | 5.919 ± 3.000 (2) |
| Hepatitis | 93.652 ± 62.000 (4) | **0.002 ± 0.000 (1)** | 0.429 ± 0.000 (3) | - | 0.217 ± 0.000 (2) |
| German | 15.399 ± 3.000 (4) | **0.005 ± 0.000 (1)** | 8.271 ± 5.000 (3) | - | 6.228 ± 4.000 (2) |
| Mushroom | 34.558 ± 12.000 (3) | **0.008 ± 0.000 (1)** | 39.021 ± 23.000 (4) | - | 3.206 ± 2.000 (2) |
| Credit Card | 38.440 ± 11.000 (3) | **0.186 ± 0.000 (1)** | 2.688 ± 1.000 (2) | - | 334.752 ± 181.000 (4) |
| Horse Colic | 45.138 ± 24.000 (4) | **0.003 ± 0.000 (1)** | 8.329 ± 4.000 (2) | - | 9.334 ± 3.000 (3) |
| Thyroid | 75.169 ± 19.000 (3) | **0.029 ± 0.000 (1)** | 85.355 ± 46.000 (4) | - | 6.492 ± 2.000 (2) |
| Spambase | 12.275 ± 3.000 (4) | **0.037 ± 0.000 (1)** | 12.083 ± 3.000 (3) | - | 5.418 ± 1.000 (2) |
| Mean Reciprocal Rank | 0.254 ± 0.043 (4) | **1.000 ± 0.000 (1)** | 0.360 ± 0.081 (3) | 0.228 ± 0.025 (5) | 0.459 ± 0.079 (2) |
| Iris | 8.969 ± 2.000 (5) | **0.002 ± 0.000 (1)** | 0.927 ± 0.000 (3) | 6.847 ± 1.000 (4) | 0.016 ± 0.000 (2) |
| Balance Scale | 32.529 ± 3.000 (5) | **0.002 ± 0.000 (1)** | 6.254 ± 2.000 (3) | 11.067 ± 5.000 (4) | 0.035 ± 0.000 (2) |
| Car | 31.089 ± 2.000 (4) | **0.002 ± 0.000 (1)** | 6.013 ± 2.000 (3) | 173.529 ± 185.000 (5) | 0.084 ± 0.000 (2) |
| Glass | 11.030 ± 1.000 (4) | **0.002 ± 0.000 (1)** | 1.628 ± 0.000 (3) | 17.268 ± 5.000 (5) | 0.201 ± 0.000 (2) |
| Contraceptive | 28.885 ± 8.000 (4) | **0.002 ± 0.000 (1)** | 4.429 ± 1.000 (3) | 23.142 ± 3.000 (4) | 0.208 ± 0.000 (2) |
| Solar Flare | 8.492 ± 3.000 (4) | **0.002 ± 0.000 (1)** | 0.245 ± 0.000 (3) | 32.101 ± 4.000 (5) | 0.206 ± 0.000 (2) |
| Wine | 29.892 ± 2.000 (4) | **0.002 ± 0.000 (1)** | 0.407 ± 0.000 (2) | 43.946 ± 27.000 (5) | 0.549 ± 0.000 (3) |
| Zoo | 101.162 ± 64.000 (4) | **0.002 ± 0.000 (1)** | 3.660 ± 1.000 (3) | - | 0.016 ± 0.000 (2) |
| Lymphography | 22.461 ± 9.000 (4) | **0.002 ± 0.000 (1)** | 0.547 ± 0.000 (3) | - | 0.349 ± 0.000 (2) |
| Segment | 36.779 ± 3.000 (4) | **0.014 ± 0.000 (1)** | 11.370 ± 2.000 (2) | - | 14.888 ± 1.000 (3) |
| Dermatology | 12.615 ± 4.000 (3) | **0.002 ± 0.000 (1)** | 6.792 ± 2.000 (2) | - | 22.681 ± 2.000 (4) |
| Landsat | 63.481 ± 7.000 (3) | **0.046 ± 0.000 (1)** | 2.285 ± 0.000 (2) | - | 871.353 ± 11.000 (4) |
| Annealing | 10.094 ± 3.000 (4) | **0.002 ± 0.000 (1)** | 2.648 ± 1.000 (3) | - | 2.857 ± 0.000 (2) |
| Splice | 16.807 ± 3.000 (3) | **0.018 ± 0.000 (1)** | 5.659 ± 2.000 (2) | - | 1818.241 ± 227.000 (4) |
| Mean Reciprocal Rank | 0.257 ± 0.044 (4) | **1.000 ± 0.000 (1)** | 0.405 ± 0.082 (3) | 0.221 ± 0.025 (5) | 0.411 ± 0.107 (2) |

Table 4: **Runtime Comparison.** We report runtime without restarts based on the optimized hyperparameters (mean ± stdev over 10 trials). We also report the ranking of each approach in brackets. The top part comprises binary classification tasks and the bottom part multi-class datasets. The datasets are sorted by the number of features. Due to scalability issues, DNDTs were only applied to datasets with a maximum of 12 features, as suggested by Yang et al. (2018).

set as suggested by the authors. However, we extended their implementation to use early stopping based on the validation loss, similar to GDTs, to reduce the runtime and prevent overfitting.

Further details can be found in the code, which we provided in the supplementary material. Furthermore, we used random restarts only for GDTs We made this decision based on the fact that CART is deterministic and therefore does not benefit from additional restarts and for GeneticTree, increasing the population size would achieve similar results. Therefore, both baseline approaches would suffer from the use of restarts, since less data can be used for the training.

| | GDT | Greedy (CART) | GeneticTree | DNDT | DL8.5 |
|---|---|---|---|---|---|
| Blood Transfusion | **0.704 ± 0.053 (1)** | 0.609 ± 0.063 (3) | 0.522 ± 0.044 (5) | 0.704 ± 0.053 (2) | 0.565 ± 0.031 (4) |
| Banknote Authentication | **0.993 ± 0.008 (1)** | 0.982 ± 0.007 (3) | 0.928 ± 0.026 (5) | 0.989 ± 0.006 (2) | 0.963 ± 0.011 (4) |
| Titanic | **0.839 ± 0.039 (1)** | 0.786 ± 0.037 (3) | 0.764 ± 0.037 (5) | 0.779 ± 0.058 (4) | 0.798 ± 0.029 (2) |
| Raisins | 0.898 ± 0.020 (2) | 0.853 ± 0.017 (4) | 0.858 ± 0.022 (3) | **0.908 ± 0.019 (1)** | 0.850 ± 0.027 (5) |
| Rice | 0.952 ± 0.009 (2) | 0.925 ± 0.004 (4) | 0.930 ± 0.005 (3) | **0.978 ± 0.003 (1)** | 0.923 ± 0.008 (5) |
| Echocardiogram | 0.575 ± 0.077 (3) | 0.560 ± 0.086 (4) | 0.603 ± 0.110 (2) | **0.758 ± 0.091 (1)** | 0.544 ± 0.084 (5) |
| Wisconsin Diagnostic Breast Cancer | 0.939 ± 0.035 (2) | 0.891 ± 0.030 (5) | 0.893 ± 0.027 (4) | **0.969 ± 0.013 (1)** | 0.895 ± 0.020 (3) |
| Loan House | **0.693 ± 0.031 (1)** | 0.688 ± 0.031 (3) | 0.691 ± 0.029 (2) | 0.607 ± 0.105 (5) | 0.688 ± 0.028 (4) |
| Heart Failure | **0.791 ± 0.041 (1)** | 0.757 ± 0.068 (2) | 0.749 ± 0.060 (3) | 0.622 ± 0.095 (5) | 0.695 ± 0.061 (4) |
| Heart Disease | **0.868 ± 0.057 (1)** | 0.760 ± 0.066 (4) | 0.801 ± 0.045 (2) | - | 0.772 ± 0.048 (3) |
| Adult | **0.893 ± 0.003 (1)** | 0.766 ± 0.008 (2) | 0.679 ± 0.015 (4) | - | 0.724 ± 0.005 (3) |
| Bank Marketing | **0.683 ± 0.011 (1)** | 0.604 ± 0.009 (2) | 0.500 ± 0.000 (4) | - | 0.585 ± 0.008 (3) |
| Cervical Cancer | 0.497 ± 0.067 (3) | **0.522 ± 0.036 (1)** | 0.500 ± 0.000 (2) | - | 0.493 ± 0.014 (4) |
| Congressional Voting | **0.974 ± 0.015 (1)** | 0.961 ± 0.019 (2) | 0.961 ± 0.019 (3) | - | 0.939 ± 0.028 (4) |
| Absenteeism | **0.677 ± 0.060 (1)** | 0.657 ± 0.039 (2) | 0.625 ± 0.037 (4) | - | 0.647 ± 0.035 (3) |
| Hepatitis | 0.668 ± 0.131 (3) | 0.670 ± 0.122 (2) | 0.600 ± 0.065 (4) | - | **0.677 ± 0.093 (1)** |
| German | **0.710 ± 0.033 (1)** | 0.621 ± 0.037 (2) | 0.544 ± 0.034 (4) | - | 0.609 ± 0.027 (3) |
| Mushroom | **1.000 ± 0.000 (1)** | **1.000 ± 0.000 (1)** | 0.967 ± 0.012 (4) | - | **1.000 ± 0.000 (1)** |
| Credit Card | **0.755 ± 0.007 (1)** | 0.653 ± 0.005 (3) | 0.646 ± 0.011 (4) | - | 0.662 ± 0.011 (2) |
| Horse Colic | **0.864 ± 0.037 (1)** | 0.816 ± 0.037 (4) | 0.849 ± 0.026 (2) | - | 0.826 ± 0.034 (3) |
| Thyroid | **0.962 ± 0.008 (1)** | 0.955 ± 0.002 (2) | 0.735 ± 0.051 (3) | - | 0.715 ± 0.007 (4) |
| Spambase | 0.899 ± 0.011 (2) | **0.916 ± 0.010 (1)** | 0.864 ± 0.016 (3) | - | 0.859 ± 0.011 (4) |
| Mean Reciprocal Rank | **0.818 ± 0.270 (1)** | 0.460 ± 0.239 (3) | 0.330 ± 0.112 (5) | 0.628 ± 0.349 (2) | 0.361 ± 0.217 (4) |
| Iris | 0.973 ± 0.022 (2) | 0.948 ± 0.025 (3) | 0.919 ± 0.042 (5) | **0.986 ± 0.014 (1)** | 0.935 ± 0.033 (4) |
| Balance Scale | 0.809 ± 0.039 (2) | 0.727 ± 0.017 (3) | 0.699 ± 0.032 (4) | **0.885 ± 0.029 (1)** | 0.662 ± 0.026 (5) |
| Car | 0.945 ± 0.021 (2) | **0.963 ± 0.010 (1)** | 0.598 ± 0.034 (5) | 0.871 ± 0.032 (3) | 0.765 ± 0.028 (4) |
| Glass | 0.721 ± 0.154 (2) | 0.697 ± 0.142 (3) | 0.606 ± 0.078 (5) | **0.764 ± 0.177 (1)** | 0.646 ± 0.111 (4) |
| Contraceptive | **0.669 ± 0.021 (1)** | 0.655 ± 0.021 (3) | 0.622 ± 0.028 (5) | 0.653 ± 0.030 (4) | 0.663 ± 0.015 (2) |
| Solar Flare | **0.521 ± 0.062 (1)** | 0.501 ± 0.003 (4) | 0.500 ± 0.000 (5) | 0.510 ± 0.030 (2) | 0.501 ± 0.002 (4) |
| Wine | **0.960 ± 0.036 (1)** | 0.934 ± 0.023 (3) | 0.927 ± 0.036 (4) | 0.951 ± 0.023 (2) | 0.890 ± 0.018 (5) |
| Zoo | 0.596 ± 0.193 (2) | **0.600 ± 0.200 (1)** | 0.569 ± 0.143 (4) | - | 0.594 ± 0.189 (3) |
| Lymphography | 0.516 ± 0.047 (2) | 0.508 ± 0.023 (4) | 0.511 ± 0.033 (3) | - | **0.516 ± 0.048 (1)** |
| Segment | **0.990 ± 0.002 (1)** | 0.980 ± 0.003 (2) | 0.874 ± 0.024 (4) | - | 0.889 ± 0.008 (3) |
| Dermatology | **0.991 ± 0.006 (1)** | 0.964 ± 0.026 (2) | 0.928 ± 0.054 (4) | - | 0.942 ± 0.028 (3) |
| Landsat | **0.968 ± 0.004 (1)** | 0.900 ± 0.006 (2) | 0.788 ± 0.022 (4) | - | 0.866 ± 0.005 (3) |
| Annealing | **0.888 ± 0.194 (1)** | 0.831 ± 0.169 (4) | 0.640 ± 0.101 (4) | - | 0.869 ± 0.186 (2) |
| Splice | **0.960 ± 0.005 (1)** | 0.936 ± 0.006 (3) | 0.759 ± 0.048 (4) | - | 0.943 ± 0.012 (2) |
| Mean Reciprocal Rank | **0.786 ± 0.247 (1)** | 0.458 ± 0.233 (3) | 0.238 ± 0.035 (5) | 0.655 ± 0.310 (2) | 0.374 ± 0.201 (4) |

Table 5: **Performance Comparison Uncertainty Quantification.** We report ROC AUC score (mean ± stdev over 10 trials). We also report the ranking of each approach in brackets. The top part comprises binary classification tasks and the bottom part multi-class datasets. The datasets are sorted by the number of features. Due to scalability issues, DNDTs were only applied to datasets with a maximum of 12 features, as suggested by Yang et al. (2018).

| | GDT | Greedy (CART) | GeneticTree | DNDT | DL8.5 |
|---|---|---|---|---|---|
| Blood Transfusion | 0.769 ± 0.037 (2) | **0.798 ± 0.015 (1)** | 0.682 ± 0.048 (5) | 0.750 ± 0.046 (4) | 0.767 ± 0.007 (3) |
| Banknote Authentication | 0.998 ± 0.002 (2) | **0.998 ± 0.001 (1)** | 0.937 ± 0.017 (5) | 0.953 ± 0.011 (4) | 0.967 ± 0.002 (3) |
| Titanic | 0.824 ± 0.014 (2) | 0.821 ± 0.008 (3) | 0.790 ± 0.015 (4) | 0.763 ± 0.051 (5) | **0.846 ± 0.007 (1)** |
| Raisins | 0.876 ± 0.009 (2) | 0.866 ± 0.004 (3) | 0.863 ± 0.004 (4) | 0.854 ± 0.019 (5) | **0.889 ± 0.004 (1)** |
| Rice | 0.927 ± 0.005 (3) | 0.929 ± 0.002 (2) | 0.922 ± 0.002 (5) | 0.926 ± 0.004 (4) | **0.931 ± 0.002 (1)** |
| Echocardiogram | 0.822 ± 0.023 (5) | 0.829 ± 0.050 (3) | 0.827 ± 0.028 (4) | 0.929 ± 0.023 (2) | **0.969 ± 0.009 (1)** |
| Wisconsin Diagnostic Breast Cancer | 0.958 ± 0.008 (3) | 0.920 ± 0.006 (4) | 0.916 ± 0.011 (5) | 0.965 ± 0.010 (2) | **0.967 ± 0.004 (1)** |
| Loan House | 0.786 ± 0.024 (3) | 0.800 ± 0.009 (2) | 0.779 ± 0.014 (4) | 0.773 ± 0.056 (5) | **0.832 ± 0.007 (1)** |
| Heart Failure | 0.853 ± 0.021 (4) | 0.878 ± 0.017 (3) | 0.827 ± 0.019 (5) | **0.942 ± 0.027 (1)** | 0.926 ± 0.008 (2) |
| Heart Disease | 0.840 ± 0.030 (4) | 0.891 ± 0.007 (2) | 0.845 ± 0.011 (3) | - | **0.914 ± 0.007 (1)** |
| Adult | 0.840 ± 0.003 (2) | **0.861 ± 0.002 (1)** | 0.793 ± 0.004 (4) | - | 0.819 ± 0.001 (3) |
| Bank Marketing | 0.862 ± 0.010 (3) | **0.882 ± 0.001 (1)** | 0.828 ± 0.001 (4) | - | 0.868 ± 0.002 (2) |
| Cervical Cancer | 0.902 ± 0.011 (3) | 0.922 ± 0.010 (2) | 0.901 ± 0.005 (4) | - | **0.932 ± 0.005 (1)** |
| Congressional Voting | 0.965 ± 0.013 (2) | 0.956 ± 0.004 (3) | 0.956 ± 0.004 (4) | - | **0.991 ± 0.002 (1)** |
| Absenteeism | **0.831 ± 0.018 (1)** | 0.806 ± 0.015 (2) | 0.677 ± 0.023 (4) | - | 0.774 ± 0.009 (3) |
| Hepatitis | 0.815 ± 0.050 (4) | 0.862 ± 0.018 (2) | 0.838 ± 0.022 (3) | - | **0.985 ± 0.008 (1)** |
| German | 0.751 ± 0.026 (3) | 0.777 ± 0.018 (2) | 0.686 ± 0.022 (4) | - | **0.791 ± 0.005 (1)** |
| Mushroom | **1.000 ± 0.000 (1)** | **1.000 ± 0.000 (1)** | 0.968 ± 0.012 (4) | - | **1.000 ± 0.000 (1)** |
| Credit Card | 0.797 ± 0.003 (3) | **0.801 ± 0.002 (1)** | 0.794 ± 0.002 (4) | - | 0.798 ± 0.002 (2) |
| Horse Colic | 0.864 ± 0.014 (4) | 0.897 ± 0.010 (2) | 0.867 ± 0.015 (3) | - | **0.944 ± 0.005 (1)** |
| Thyroid | 0.947 ± 0.006 (2) | **0.987 ± 0.001 (1)** | 0.833 ± 0.034 (3) | - | 0.805 ± 0.003 (4) |
| Spambase | 0.882 ± 0.008 (3) | **0.966 ± 0.002 (1)** | 0.874 ± 0.017 (4) | - | 0.886 ± 0.003 (2) |
| Mean Reciprocal Rank | 0.426 ± 0.207 (3) | 0.633 ± 0.287 (2) | 0.258 ± 0.046 (5) | 0.372 ± 0.250 (4) | **0.754 ± 0.302 (1)** |
| Iris | 0.964 ± 0.016 (2) | 0.964 ± 0.009 (3) | 0.939 ± 0.013 (5) | 0.960 ± 0.014 (4) | **0.996 ± 0.004 (1)** |
| Balance Scale | 0.819 ± 0.025 (2) | **1.000 ± 0.000 (1)** | 0.778 ± 0.017 (4) | 0.808 ± 0.018 (3) | 0.768 ± 0.007 (5) |
| Car | 0.868 ± 0.037 (2) | **0.995 ± 0.001 (1)** | 0.720 ± 0.049 (5) | 0.725 ± 0.036 (4) | 0.838 ± 0.007 (3) |
| Glass | 0.773 ± 0.037 (4) | **0.995 ± 0.006 (1)** | 0.593 ± 0.038 (5) | 0.803 ± 0.067 (2) | 0.785 ± 0.011 (3) |
| Contraceptive | 0.545 ± 0.019 (4) | 0.578 ± 0.008 (2) | 0.509 ± 0.032 (5) | 0.555 ± 0.024 (3) | **0.585 ± 0.004 (1)** |
| Solar Flare | 0.770 ± 0.012 (4) | **0.825 ± 0.007 (1)** | 0.766 ± 0.006 (5) | 0.792 ± 0.018 (3) | 0.804 ± 0.006 (2) |
| Wine | 0.990 ± 0.011 (3) | **1.000 ± 0.000 (1)** | 0.947 ± 0.015 (5) | 0.998 ± 0.004 (2) | 0.976 ± 0.005 (4) |
| Zoo | 0.978 ± 0.011 (3) | **1.000 ± 0.000 (1)** | 0.880 ± 0.064 (4) | - | **1.000 ± 0.000 (1)** |
| Lymphography | 0.905 ± 0.038 (3) | **0.993 ± 0.012 (1)** | 0.816 ± 0.044 (4) | - | 0.969 ± 0.008 (2) |
| Segment | 0.956 ± 0.009 (2) | **0.997 ± 0.002 (1)** | 0.763 ± 0.046 (4) | - | 0.799 ± 0.004 (3) |
| Dermatology | 0.974 ± 0.019 (2) | **0.988 ± 0.003 (1)** | 0.900 ± 0.064 (4) | - | 0.960 ± 0.005 (3) |
| Landsat | 0.881 ± 0.007 (2) | **0.909 ± 0.002 (1)** | 0.676 ± 0.044 (4) | - | 0.816 ± 0.003 (3) |
| Annealing | 0.981 ± 0.005 (3) | 0.987 ± 0.004 (2) | 0.842 ± 0.062 (4) | - | **0.996 ± 0.001 (1)** |
| Splice | 0.888 ± 0.009 (3) | **0.957 ± 0.003 (1)** | 0.670 ± 0.068 (4) | - | 0.925 ± 0.003 (2) |
| Mean Reciprocal Rank | 0.387 ± 0.103 (3) | **0.881 ± 0.231 (1)** | 0.229 ± 0.025 (5) | 0.357 ± 0.097 (4) | 0.544 ± 0.301 (2) |

Table 6: **Train Performance Comparison.** We report F1-score (mean ± stdev over 10 trials) on the training data. We also report the ranking of each approach for the corresponding dataset in brackets. The top part comprises binary classification tasks and the bottom part multi-class datasets. The datasets are sorted based on the number of features. Due to scalability issues, DNDTs were only applied to datasets with a maximum of 12 features, as suggested by Yang et al. (2018).

| Dataset Name | Number of Features | Number of Samples | Samples by Class | Number of Classes | Link |
|---|---|---|---|---|---|
| Blood Transfusion | 4 | 748 | 178 / 570 | 2 | https://archive.ics.uci.edu/ml/datasets/Blood+Transfusion+Service+Center |
| Banknote Authentication | 4 | 1372 | 610 / 762 | 2 | https://archive.ics.uci.edu/ml/datasets/banknote+authentication |
| Titanic | 7 | 891 | 342 / 549 | 2 | https://www.kaggle.com/c/titanic |
| Raisin | 7 | 900 | 450 /450 | 2 | https://archive.ics.uci.edu/ml/datasets/Raisin+Dataset |
| Rice | 7 | 3810 | 1630 / 2180 | 2 | https://archive.ics.uci.edu/ml/datasets/Rice+%28Cammeo+and+Osmancik%29 |
| Echocardiogram | 8 | 132 | 107 / 25 | 2 | https://archive.ics.uci.edu/ml/datasets/echocardiogram |
| Wisconsin Diagnostic Breast Cancer | 10 | 569 | 212 / 357 | 2 | https://archive.ics.uci.edu/ml/datasets/breast+cancer+wisconsin+(diagnostic) |
| Loan House | 11 | 614 | 422 / 192 | 2 | https://www.kaggle.com/code/sazid28/home-loan-prediction/data |
| Heart Failure | 12 | 299 | 96 /203 | 2 | https://archive.ics.uci.edu/ml/datasets/Heart+failure+clinical+records |
| Heart Disease | 13 | 303 | 164 / 139 | 2 | https://archive.ics.uci.edu/ml/datasets/heart+disease |
| Adult | 14 | 32561 | 7841 / 24720 | 2 | https://archive.ics.uci.edu/ml/datasets/adult |
| Bank Marketing | 14 | 45211 | 5289 / 39922 | 2 | https://archive.ics.uci.edu/ml/datasets/bank+marketing |
| Cervical Cancer | 15 | 858 | 55 / 803 | 2 | https://archive.ics.uci.edu/ml/datasets/Cervical+cancer+%28Risk+Factors%29 |
| Congressional Voting | 16 | 435 | 267 / 168 | 2 | https://archive.ics.uci.edu/ml/datasets/congressional+voting+records |
| Absenteeism | 18 | 740 | 279 / 461 | 2 | https://archive.ics.uci.edu/ml/datasets/Absenteeism+at+work |
| Hepatitis | 19 | 155 | 32 / 123 | 2 | https://archive.ics.uci.edu/ml/datasets/hepatitis |
| German | 20 | 1000 | 300 / 700 | 2 | https://archive.ics.uci.edu/ml/datasets/statlog+(german+credit+data) |
| Mushrooms | 22 | 8124 | 4208 /3916 | 2 | https://archive.ics.uci.edu/ml/datasets/mushroom |
| Credit Card | 23 | 30000 | 23364 / 6636 | 2 | https://archive.ics.uci.edu/ml/datasets/default+of+credit+card+clients |
| Horse Colic | 27 | 368 | 232 / 136 | 2 | https://archive.ics.uci.edu/ml/datasets/Horse+Colic |
| Thyroid | 29 | 9172 | 2401 / 6771 | 2 | https://archive.ics.uci.edu/ml/datasets/thyroid+disease |
| Spambase | 57 | 4601 | 1813 / 2788 | 2 | https://archive.ics.uci.edu/ml/datasets/spambase |
| Iris | 4 | 150 | 50 / 50 / 50 | 3 | https://archive.ics.uci.edu/ml/datasets/iris |
| Balance Scale | 4 | 625 | 49 / 288 / 288 | 3 | https://archive.ics.uci.edu/ml/datasets/balance+scale |
| Car | 6 | 1728 | 384 / 69 / 1210 / 65 | 4 | https://archive.ics.uci.edu/ml/datasets/car+evaluation |
| Glass | 9 | 214 | 70 / 76 / 17 / 13 / 9 / 29 | 6 | https://archive.ics.uci.edu/ml/datasets/glass+identification |
| Contraceptive | 9 | 1473 | 629 / 333 / 511 | 3 | https://archive.ics.uci.edu/ml/datasets/Contraceptive+Method+Choice |
| Solar Flare | 10 | 1389 | 1171 / 141 / 40 / 20 / 9 / 4 / 3 / 1 | 8 | http://archive.ics.uci.edu/ml/datasets/solar+flare |
| Wine | 12 | 178 | 59 / 71 / 48 | 3 | https://archive.ics.uci.edu/ml/datasets/wine |
| Zoo | 16 | 101 | 41 / 20 / 5 / 13 / 4 / 8 /10 | 7 | https://archive.ics.uci.edu/ml/datasets/zoo |
| Lymphography | 18 | 148 | 2 / 81 /61 /4 | 4 | https://archive.ics.uci.edu/ml/datasets/Lymphography |
| Segment | 19 | 2310 | 330 / 330 / 330 / 330 / 330 / 330 / 330 | 7 | https://archive.ics.uci.edu/ml/datasets/image+segmentation |
| Dermatology | 34 | 366 | 112 / 61 / 72 / 49 / 52 / 20 | 6 | https://archive.ics.uci.edu/ml/datasets/dermatology |
| Landsat | 36 | 6435 | 1533 / 703 / 1358 / 626 / 707 /1508 | 6 | https://archive.ics.uci.edu/ml/datasets/Statlog+(Landsat+Satellite) |
| Annealing | 38 | 798 | 8 / 88 / 608 / 60 / 34 | 5 | https://archive.ics.uci.edu/ml/datasets/Annealing |
| Splice | 60 | 3190 | 767 / 768 / 1655 | 3 | https://archive.ics.uci.edu/ml/datasets/Molecular+Biology+(Splice-junction+Gene+Sequences) |

Table 7: **Dataset Specifications.**

| Dataset Name | depth | lr_index | lr_values | lr_leaf | optimizer | batch_size | epochs | restarts | early_stopping_epochs |
|---|---|---|---|---|---|---|---|---|---|
| Blood Transfusion | 7 | 0.1000 | 0.1000 | 0.0050 | adam | 512 | 10000 | 10 | 200 |
| Banknote Authentication | 11 | 0.0100 | 0.0100 | 0.0500 | adam | 512 | 10000 | 10 | 200 |
| Titanic | 7 | 0.0100 | 0.0500 | 0.0050 | adam | 512 | 10000 | 10 | 200 |
| Raisins | 11 | 0.0500 | 0.0010 | 0.0100 | adam | 512 | 10000 | 10 | 200 |
| Rice | 9 | 0.0100 | 0.0005 | 0.0005 | adam | 512 | 10000 | 10 | 200 |
| Echocardiogram | 11 | 0.0050 | 0.1000 | 0.0010 | adam | 512 | 10000 | 10 | 200 |
| Wisconsin Diagnostic Breast Cancer | 9 | 0.0001 | 0.0005 | 0.0050 | adam | 512 | 10000 | 10 | 200 |
| Loan House | 9 | 0.0500 | 0.0500 | 0.0005 | adam | 512 | 10000 | 10 | 200 |
| Heart Failure | 9 | 0.0050 | 0.1000 | 0.0050 | adam | 512 | 10000 | 10 | 200 |
| Heart Disease | 5 | 0.0500 | 0.0005 | 0.0100 | adam | 512 | 10000 | 10 | 200 |
| Adult | 11 | 0.0500 | 0.0100 | 0.0050 | adam | 512 | 10000 | 10 | 200 |
| Bank Marketing | 3 | 0.1000 | 0.0001 | 0.0010 | adam | 512 | 10000 | 10 | 200 |
| Cervical Cancer | 3 | 0.0500 | 0.0001 | 0.0500 | adam | 512 | 10000 | 10 | 200 |
| Congressional Voting | 11 | 0.1000 | 0.0010 | 0.0005 | adam | 512 | 10000 | 10 | 200 |
| Absenteeism | 11 | 0.0500 | 0.0010 | 0.0100 | adam | 512 | 10000 | 10 | 200 |
| Hepatitis | 3 | 0.0001 | 0.1000 | 0.0001 | adam | 512 | 10000 | 10 | 200 |
| German | 7 | 0.0010 | 0.0100 | 0.0050 | adam | 512 | 10000 | 10 | 200 |
| Mushroom | 9 | 0.0050 | 0.0100 | 0.0500 | adam | 512 | 10000 | 10 | 200 |
| Credit Card | 7 | 0.0010 | 0.0100 | 0.0010 | adam | 512 | 10000 | 10 | 200 |
| Horse Colic | 5 | 0.0500 | 0.0005 | 0.0005 | adam | 512 | 10000 | 10 | 200 |
| Thyroid | 11 | 0.0050 | 0.0100 | 0.0050 | adam | 512 | 10000 | 10 | 200 |
| Spambase | 3 | 0.0100 | 0.0100 | 0.0100 | adam | 512 | 10000 | 10 | 200 |
| Iris | 7 | 0.0500 | 0.1000 | 0.0500 | adam | 512 | 10000 | 10 | 200 |
| Balance Scale | 11 | 0.0005 | 0.0050 | 0.0050 | adam | 512 | 10000 | 10 | 200 |
| Car | 11 | 0.0100 | 0.0100 | 0.0500 | adam | 512 | 10000 | 10 | 200 |
| Glass | 9 | 0.1000 | 0.0010 | 0.1000 | adam | 512 | 10000 | 10 | 200 |
| Contraceptive | 9 | 0.1000 | 0.0050 | 0.0010 | adam | 512 | 10000 | 10 | 200 |
| Solar Flare | 7 | 0.0001 | 0.0050 | 0.1000 | adam | 512 | 10000 | 10 | 200 |
| Wine | 11 | 0.0500 | 0.1000 | 0.0500 | adam | 512 | 10000 | 10 | 200 |
| Zoo | 7 | 0.0100 | 0.0005 | 0.0010 | adam | 512 | 10000 | 10 | 200 |
| Lymphography | 9 | 0.0100 | 0.0005 | 0.0010 | adam | 512 | 10000 | 10 | 200 |
| Segment | 11 | 0.1000 | 0.0500 | 0.0500 | adam | 512 | 10000 | 10 | 200 |
| Dermatology | 9 | 0.0100 | 0.0100 | 0.0500 | adam | 512 | 10000 | 10 | 200 |
| Landsat | 11 | 0.0100 | 0.0100 | 0.0500 | adam | 512 | 10000 | 10 | 200 |
| Annealing | 7 | 0.0500 | 0.1000 | 0.0500 | adam | 512 | 10000 | 10 | 200 |
| Splice | 9 | 0.0500 | 0.0010 | 0.1000 | adam | 512 | 10000 | 10 | 200 |

Table 8: **GDT Hyperparameters**

| Dataset Name | max_depth | criterion | max_features | min_impurity_ decrease | min_impurity_ split | min_samples_ leaf | min_samples_ split | ccp_alpha |
|---|---|---|---|---|---|---|---|---|
| Blood Transfusion | 5 | entropy | None | 0.0 | None | 1 | 10 | 0.0 |
| Banknote Authentication | 7 | gini | None | 0.0 | None | 1 | 10 | 0.0 |
| Titanic | 3 | entropy | None | 0.0 | None | 5 | 5 | 0.0 |
| Raisins | 3 | entropy | None | 0.0 | None | 50 | 2 | 0.0 |
| Rice | 3 | gini | None | 0.0 | None | 5 | 2 | 0.0 |
| Echocardiogram | 3 | gini | None | 0.0 | None | 1 | 50 | 0.0 |
| Wisconsin Diagnostic Breast Cancer | 5 | gini | None | 0.0 | None | 50 | 50 | 0.0 |
| Loan House | 3 | gini | None | 0.0 | None | 1 | 50 | 0.0 |
| Heart Failure | 3 | gini | None | 0.0 | None | 5 | 2 | 0.0 |
| Heart Disease | 7 | gini | None | 0.0 | None | 5 | 2 | 0.0 |
| Adult | 11 | gini | None | 0.0 | None | 10 | 50 | 0.0 |
| Bank Marketing | 11 | entropy | None | 0.0 | None | 5 | 50 | 0.0 |
| Cervical Cancer | 5 | entropy | None | 0.0 | None | 5 | 2 | 0.0 |
| Congressional Voting | 3 | gini | None | 0.0 | None | 50 | 2 | 0.0 |
| Absenteeism | 7 | gini | None | 0.0 | None | 1 | 10 | 0.0 |
| Hepatitis | 3 | gini | None | 0.0 | None | 1 | 50 | 0.0 |
| German | 5 | entropy | None | 0.0 | None | 5 | 2 | 0.0 |
| Mushroom | 9 | gini | None | 0.0 | None | 1 | 2 | 0.0 |
| Credit Card | 5 | entropy | None | 0.0 | None | 10 | 50 | 0.0 |
| Horse Colic | 5 | entropy | None | 0.0 | None | 10 | 2 | 0.0 |
| Thyroid | 11 | gini | None | 0.0 | None | 1 | 10 | 0.0 |
| Spambase | 11 | entropy | None | 0.0 | None | 1 | 10 | 0.0 |
| Iris | 3 | gini | None | 0.0 | None | 1 | 50 | 0.0 |
| Balance Scale | 11 | gini | None | 0.0 | None | 1 | 2 | 0.0 |
| Car | 11 | entropy | None | 0.0 | None | 1 | 2 | 0.0 |
| Glass | 11 | gini | None | 0.0 | None | 1 | 2 | 0.0 |
| Contraceptive | 5 | entropy | None | 0.0 | None | 10 | 50 | 0.0 |
| Solar Flare | 9 | gini | None | 0.0 | None | 1 | 5 | 0.0 |
| Wine | 5 | gini | None | 0.0 | None | 1 | 5 | 0.0 |
| Zoo | 7 | gini | None | 0.0 | None | 1 | 2 | 0.0 |
| Lymphography | 9 | entropy | None | 0.0 | None | 1 | 2 | 0.0 |
| Segment | 11 | entropy | None | 0.0 | None | 1 | 2 | 0.0 |
| Dermatology | 7 | entropy | None | 0.0 | None | 1 | 5 | 0.0 |
| Landsat | 9 | entropy | None | 0.0 | None | 10 | 2 | 0.0 |
| Annealing | 5 | gini | None | 0.0 | None | 1 | 2 | 0.0 |
| Splice | 9 | gini | None | 0.0 | None | 5 | 2 | 0.0 |

Table 9: **CART Hyperparameters**

| Dataset Name | n_thresholds | n_trees | max_iter | cross_prob | mutation_prob |
|---|---|---|---|---|---|
| Blood Transfusion | 10 | 250 | 100 | 1.0 | 0.5 |
| Banknote Authentication | 10 | 50 | 500 | 0.4 | 0.9 |
| Titanic | 10 | 500 | 250 | 0.6 | 0.8 |
| Raisins | 10 | 100 | 50 | 1.0 | 0.6 |
| Rice | 10 | 100 | 250 | 0.4 | 0.3 |
| Echocardiogram | 10 | 500 | 50 | 0.0 | 0.7 |
| Wisconsin Diagnostic Breast Cancer | 10 | 250 | 500 | 0.4 | 0.3 |
| Loan House | 10 | 100 | 500 | 1.0 | 0.5 |
| Heart Failure | 10 | 100 | 10 | 0.4 | 0.7 |
| Heart Disease | 10 | 500 | 250 | 1.0 | 0.0 |
| Adult | 10 | 250 | 100 | 0.2 | 0.7 |
| Bank Marketing | 10 | 50 | 50 | 0.4 | 0.1 |
| Cervical Cancer | 10 | 50 | 50 | 0.4 | 0.1 |
| Congressional Voting | 10 | 50 | 50 | 0.4 | 0.1 |
| Absenteeism | 10 | 500 | 250 | 0.2 | 0.5 |
| Hepatitis | 10 | 100 | 100 | 0.2 | 0.8 |
| German | 10 | 500 | 250 | 0.2 | 0.9 |
| Mushroom | 10 | 500 | 250 | 0.8 | 0.6 |
| Credit Card | 10 | 50 | 10 | 0.6 | 0.8 |
| Horse Colic | 10 | 500 | 250 | 0.6 | 0.8 |
| Thyroid | 10 | 500 | 250 | 1.0 | 0.5 |
| Spambase | 10 | 100 | 500 | 0.8 | 0.9 |
| Iris | 10 | 100 | 250 | 1.0 | 0.1 |
| Balance Scale | 10 | 250 | 500 | 0.8 | 0.4 |
| Car | 10 | 250 | 500 | 0.6 | 0.4 |
| Glass | 10 | 250 | 500 | 0.2 | 0.7 |
| Contraceptive | 10 | 250 | 250 | 0.4 | 0.2 |
| Solar Flare | 10 | 50 | 50 | 0.4 | 0.1 |
| Wine | 10 | 100 | 100 | 0.6 | 0.3 |
| Zoo | 10 | 500 | 250 | 0.2 | 0.8 |
| Lymphography | 10 | 50 | 500 | 0.4 | 0.9 |
| Segment | 10 | 500 | 500 | 0.2 | 0.7 |
| Dermatology | 10 | 500 | 500 | 0.2 | 0.8 |
| Landsat | 10 | 50 | 250 | 0.2 | 0.8 |
| Annealing | 10 | 250 | 500 | 0.2 | 0.7 |
| Splice | 10 | 100 | 500 | 0.6 | 0.8 |

Table 10: **GeneticTree Hyperparameters**

