# OpenReview forum: "Learning Axis-Aligned Decision Trees with Gradient Descent"
_ICLR.cc/2023/Conference — Submitted to ICLR 2023_

### Official Review · Reviewer_ePjz · 2022-10-23

**Confidence:** 5
**Correctness:** 3
**Technical Novelty And Significance:** 3
**Empirical Novelty And Significance:** 3
**Recommendation:** 6

**Clarity, Quality, Novelty And Reproducibility:**

The **clarity** and **quality** of the work should be improved. The method is described in great detail and the contributions are clearly stated, but some concepts are badly named/referenced, the comparison is restricted to very few baselines without a strong motivation.

The tree routing formulation is **novel** to the best of my knowledge. To highlight this contribution, a discussion on the difference with existing formulations should be reported. The optimization procedure itself is not novel: it boils down to a well-known trick in discrete optimization.

The **reproducibility** of the work is very high. The implementation and scripts for running the experiments were submitted as supplementary material, the method is described in great level of detail and the hyper-parameter values are reported for all datasets and methods.

**Strength And Weaknesses:**

## Strengths

The formulation and method are described in great level of detail. The contributions are clearly stated from the beginning and the empirical analysis studies several aspects of the method's performance. Of particular interest is the study of the performance of all methods using the default values for the hyper-parameters, which shows that all studied methods are quite robust to non-optimal choices of hyper-parameters.

To the best of my knowledge, the proposed formulation of tree routing is novel. It is also easier to read and understand than those of existing methods, in particular of the cited work by [Bertsimas and Dunn, 2017] and of [3], and more practical, as it does not require constraints reflecting the hierarchy of the tree.


## Weaknesses
1. **The motivation for learning axis-aligned splits (as opposed to oblique) is weak**. In the paper it is said that axis-aligned trees are more interpretable than oblique ones. However one could argue that learning oblique splits allow to obtain trees shallower than with axis-aligned ones, hence with fewer decision rules. For this, trees with oblique splits could be easier to read and to interpret. Moreover, the proposed approach learns fully-grown trees, which are considered less interpretable than sparse ones (e.g., see discussion in [5]).

It would be valuable to develop this discussion in the paper, especially because:
- the argument that axis-aligned trees are more interpretable is used to justify the choice of restricting the empirical comparison only to axis-aligned tree baselines;
- the actual interpretability of the learned trees is not studied in the paper.

 2. **The work should be better placed within the relevant literature**.

- Some references are missing from the literature on optimal decision trees and the one on soft-trees, e.g. [2], [3], [4]. In particular, [3] also optimizes deterministic tree routings by gradient descent, hence it is possibly the closest related work. It would be valuable to discuss the differences between the two methods' formulations (e.g., see "Strengths") and of optimization choices (e.g., in which operations the biases are introduced, output nature, loss options). The current work could also take inspiration from the pruning procedure of [3] for extending the method to learning of sparse trees.

- Several times in the main text the optimization procedure is described as "an adjusted gradient flow". However,  the updates are discrete so it is not clear why the procedure is described as a "flow". Also, the term "straight-through operator" should be preferred over "adjusted" because it is more specific and references the rich literature that leverages it, starting from the seminal work [1] that should be cited in the paper. Finally, it should be highlighted in the paper that the "straight-through operator" introduces a mismatch between forward and backward passes, hence the **claim that "GDTs directly optimize the loss function" (page 2) is not technically true**.

- In page 6 an algorithm for performing gradient descent with momentum is mentioned but its name or reference is not provided.

- The paper should be self-contained. However in Section 4 the results of four experiments are commented in the main text but reported in the appendix. I suggest to choose the most significant experiment to report and comment in the main text, and defer the remaining in the appendix. To gain some space, Sections 3.3 and 3.4 could be considerably compacted (e.g., Eq 7 and 8 are not needed once the term straight-through operator is used).

3. To judge the potential impact of the method, the performance of existing differentiable tree learning methods and, when feasible, of optimal tree learning methods should be reported as well. The current empirical evaluation reports the results of baselines that are not always state-of-the-art in terms of accuracy, and it would be valuable to check whether the proposed method improve over the running times of these lines of works, even though it does not over greedy approaches.

## Minors
- page 3: $d$ is not defined

[1] Yoshua Bengio, Nicholas Léonard, Aaron C. Courville: Estimating or Propagating Gradients Through Stochastic Neurons for Conditional Computation, 2013

[2] Ryutaro Tanno, Kai Arulkumaran, Daniel C. Alexander, Antonio Criminisi, Aditya V. Nori: Adaptive Neural Trees. ICML 2019

[3] Valentina Zantedeschi, Matt J. Kusner, Vlad Niculae: Learning Binary Decision Trees by Argmin Differentiation. ICML 2021

[4] Emir Demirovic, Anna Lukina, Emmanuel Hebrard, Jeffrey Chan, James Bailey, Christopher Leckie, Kotagiri Ramamohanarao, Peter J. Stuckey: MurTree: Optimal Decision Trees via Dynamic Programming and Search. J. Mach. Learn. Res., 2022.

[5] Cynthia Rudin, Chaofan Chen, Zhi Chen, Haiyang Huang, Lesia Semenova, Chudi Zhong: Interpretable Machine Learning: Fundamental Principles and 10 Grand Challenges, Statistics Surveys, 2022


**Summary Of The Paper:**

The paper proposes a method for learning fully-grown binary decision trees with axis-aligned splits by gradient descent.
For that, it first describes a formulation for determining a point route through the tree (based on an integer formulation of logical-ands and logical-ors) given the set of decision splits, and its output. The splits are parametrized by an index vector (that selects the features to evaluate at each branch node) and a threshold vector (that determines whether the point should go left or right), and the leaf nodes are each associated by a probability distribution over possible outputs.

Then, the paper describes how to optimize the tree parameters. As the overall decision function is piece-wise constant, the authors make use of the straight-through operator to update the discrete parameters by gradient descent.
An empirical analysis assesses how the proposed method compares with CART and a genetic tree search method in terms of test accuracies, generalization gap, running times, prediction confidence, robustness to hyper-parameter changes.

**Summary Of The Review:**

Overall the paper proposes a novel formulation for learning fully-grown axis-aligned decision trees. The presentation should be improved, starting from better placing the work within the relevant literature, providing a thorough discussion of the benefits of learning axis-aligned splits and including (at least) an optimal tree baseline in the empirical comparison. Also the misleading claim of page 2 (see Weaknesses) should be rectified. I would be happy to increase my score accordingly.

**update after rebuttal**
The authors thoroughly revised the paper based on the feedback, and for this I raised my score.

The only remaining issue from my side is the motivation for learning axis-aligned split functions and for restricting the empirical comparison only to this type of trees. The interpretability of the learned trees (which is used to justify axis-aligned functions) should have been analyzed, for instance by comparing the number of decision functions needed by each method to achieve similar accuracies. Without such analysis, it is not clear why a practitioner would prefer this new method to existing ones, in particular as it learns only fully-grown trees.

---

> ### Author Response · Authors · 2022-11-18
> **In response to your encouraging and constructive feedback (2/2)**
>
>
> ##### Placement in relevant literature
> > Some references are missing from the literature on optimal decision trees and the one on soft-trees, e.g. [2], [3], [4]. In particular, [3] also optimizes deterministic tree routings by gradient descent, hence it is possibly the closest related work.
>
> * [2] Ryutaro et al. (2019) Thanks for highlighting this work. While we believe that our approach differs from ANTs significantly, we adjusted our related work section accordingly and highlighted the relevant differences: ANTs employ a stochastic routing based on a Bernoulli distribution which the mean as learned parameter, while the routing in our approach is deterministic. Furthermore, ANTs utilize one or more non-linear transformer modules at the edges. As a result, ANTs are, in contrast to our approach, not axis-aligned.
> * [3] Zantedeschi et al. (2021) require a differentiable split function for their framework. More specifically, during their experiments, they use a linear split function implemented as a neural network with a single dense layer where the number of neurons equals the number of features. Accordingly, the trees are oblique opposed to axis aligned trees, which is our focus. Still, it is related to our work, especially through the tree routing and therefore, we have included it in our related work section.
> * [4] Demirovic et al. (2022): We added MurTree to the discussion in our related work section and implemented PyDL8.5 as a benchmark for the evaluation as an optimized version of DL8.5 which now includes several improvements from MurTree. The main drawback of current optimal decision tree approaches which still is not addressed is the need for binary features. This requires discretization and therefore comes with additional challenges and design choices. Our results show that GDTs outperforms DL8.5, especially on binary classification tasks.
>
> > The current work could also take inspiration from the pruning procedure of [3] for extending the method to learning of sparse trees.
>
> Thank you very much for pointing this out. We also think that this can be an inspiration for implementing a pruning mechanism with our approach and we will address this in future work.
>
> > Also, the term "straight-through operator" should be preferred over "adjusted" because it is more specific and references the rich literature that leverages it, starting from the seminal work [1] that should be cited in the paper.
>
> Thanks for pointing out the seminal work of [1] Bengio et al. (2013). We clarified the formulation using the term “straight-through operator” as you suggested and cited [1] accordingly. We further reduced section 3.3 including Eq. 7 and 8 as suggested.
>
> > The paper should be self-contained. However in Section 4 the results of four experiments are commented in the main text but reported in the appendix.
>
> The most important experimental results are already in the main part of the paper. Due to lack of space, even after reducing the size of Section 3.3 as suggested, we had to move the tables comprising further results in the appendix. However, we added an additional table summarizing most results listed in the appendix to the main part.
>
> ##### Empirical Evaluation
> > To judge the potential impact of the method, the performance of existing differentiable tree learning methods and, when feasible, of optimal tree learning methods should be reported as well.
>
> We added two additional benchmarks to the evaluation, namely Deep Neural Decision Trees (DNDT) as a related approach to learning DTs with gradient descent and Optimal Decision Trees (DL8.5 with the improvements from MurTree). We were able to show that GDTs outperform both new baselines, especially on binary classification tasks.

---

> ### Author Response · Authors · 2022-11-18
> **In response to your encouraging and constructive feedback (1/2)**
>
> We appreciate the reviewer's constructive feedback and further thank the reviewer for highlighting the strengths and novelty of our approach. We have already adjusted our paper in the uploaded, revised version based on your feedback. A markup version highlighting the relevant text changes can additionally be found in the supplementary material. In the following, we address each point of your review:
>
> ##### Motivation against oblique splits
> > The motivation for learning axis-aligned splits (as opposed to oblique) is weak. In the paper it is said that axis-aligned trees are more interpretable than oblique ones. However, one could argue that learning oblique splits allow to obtain trees shallower than with axis-aligned ones, hence with fewer decision rules. For this, trees with oblique splits could be easier to read and to interpret.
>
> We acknowledge that this should not be formulated as a general statement, since shallow oblique trees might be more interpretable compared to certain axis-aligned trees. However, especially with an increasing number of features, it becomes difficult to even comprehend a single split for tabular data (if not sparse with respect to the features). This is also supported by Molnar (2020) where the authors argue that humans can't comprehend explanations considering more than three features at the time. We also clarified this in the revised version of the paper accordingly.
> Molnar, C. (2020). Interpretable machine learning. Lulu.com.
>
>
> > Moreover, the proposed approach learns fully-grown trees, which are considered less interpretable than sparse ones (e.g., see discussion in [5]).
>
> We agree that learning sparse trees would increase the interpretability even further. Therefore, we want to address pruning mechanisms for our approach in future work.

---

### Official Review · Reviewer_tioQ · 2022-10-24

**Confidence:** 5
**Correctness:** 2
**Technical Novelty And Significance:** 2
**Empirical Novelty And Significance:** 2
**Recommendation:** 3

**Clarity, Quality, Novelty And Reproducibility:**

I have concerns regarding the novelty with respect to previous approaches that also aim to train trees with gradient-based optimisation. There is an existing body of work that follows the same high-level strategy as this paper (i.e., constructing a tree with soft splits), with perhaps the most relevant piece of work being Yang et al. (2018). In addition, I think the quality of the paper could be improved by running some more experiments that back up some of the claims.

**Strength And Weaknesses:**

### Strengths
* Investigating alternatives to greedy splitting for constructing decision trees is an interesting problem, and the paper does a good job of highlighting why this is the case by including an example of where greedy splitting is not optimal.
* The method is quite simple and the description is relatively easy to follow.

## Weaknesses
* This paper is not the first to use gradient-based optimisation for axis-aligned decision tree induction. For example, see work by Gouk et al. (2019), Yang et al. (2018), and Suarez and Lutsko (1999). The paper should include a more thorough discussion of relevant related work.
* The authors assume that gradient descent performs global optimisation, but this is in general only true for some classes of functions. The authors should demonstrate that their decision tree representation belongs to one of these function classes if they want to make that claim.
* Experimental results are quite limited. A comparison should be performed with more recent prior work (e.g., those listed above), and a neural network baseline should also be included. It would also be interesting if the paper included experiments on something beyond just classification accuracy on UCI-style tabular datasets; e.g., investigation of performance on regression datasets, or other tasks where the ability to specify an arbitrary loss function would be useful. The authors may also consider evaluating how well these decision trees are able to leverage standard ensembling techniques---something that standard decision trees are excellent candidates for.

1. Gouk et al. Stochastic Gradient Trees. Asian Conference on Machine Learning, 2019.
2. Yang et al. Deep Neural Decision Trees. ICML Workshop on Human Interpretability in Machine Learning, 2018.
3. Suarez and Lutsko. Globally Optimal Fuzzy Decision Trees for Classificaiton and Regression. IEEE TPAMI, 1999.


**Summary Of The Paper:**

The paper proposes a method for constructing decision tree classifiers that does not rely on performing greedy splits. Instead, the method constructs a soft decision tree approximation to conventional hard split axis-aligned decision trees, and then trains this soft tree with backpropagation and gradient-based optimisation algorithms. The performance of the algorithm is compared with conventional CART trees and an evolutionary optimisation induction algorithm on a variety of small tabular datasets. The proposed method had comparable performance with CART.

**Summary Of The Review:**

Given the lack of acknowledgement and comparison with very related prior work, missing baselines, and minimally informative experimental results, I cannot recommend acceptance. I would also argue that since there is no representation learning in this paper, that ICLR is perhaps the wrong venue.

---

> ### Author Response · Authors · 2022-11-18
> **Response to your constrictive feedback**
>
> We thank the reviewer for their helpful feedback, which emphasizes details that we clarify in our revised paper which is already uploaded. A markup version highlighting the relevant text changes can additionally be found in the supplementary material. Please see our responses to your specific concerns below:
>
> > This paper is not the first to use gradient-based optimization for axis-aligned decision tree induction. For example, see work by Gouk et al. (2019), Yang et al. (2018), and Suarez and Lutsko (1999). The paper should include a more thorough discussion of relevant related work.
>
> Thank you for pointing this out. We acknowledge that a more thorough discussion of related work was necessary, and we addressed this in our revised paper. We further added DNDTs and an optimal DT (DL8.5) baseline to our evaluation. We were able to show that GDTs outperform both new baselines, especially on binary classification tasks.
> * Gouk et al. (2019) propose stochastic gradient trees (SGTs) which use stochastic gradient information as source of supervision to incrementally learn a decision tree. This is substantially different from our approach, as SGTs do not apply gradient descent as it is the case for GDTs but evaluate the loss function for each possible split to find the one that yields to the maximum loss reduction.
> * Suarez and Lutsko (1999) relax the hard splits derived from an already existing DT (e.g., learned using CART) and optimizes the soft splits with backpropagation. However, they are not able to *learn* a DT since they require an already trained tree as input.
> * Yang et al. (2018) propose DNDTs which realize tree models as neural networks, utilizing a soft binning function for splitting. Therefore, the resulting trees are soft (even though the authors suggest using the straight-through operator similar to our work if hard decisions are desired) and axis-aligned, which makes this work closely related to our approach. Besides a discussion of the differences between DNDTs and our approach, we also included DNDTs in our empirical evaluation. In contrast to our approach, DNDTs scale poorly with the number of features due to the use of the Kronecker product, which allows an application of single trees only to datasets with at most 12 features.
>
> > The authors assume that gradient descent performs global optimization, but this is in general only true for some classes of functions. The authors should demonstrate that their decision tree representation belongs to one of these function classes if they want to make that claim.
>
> With the expression “global optimization” we wanted to emphasize that our gradient-based approach optimizes all tree parameters simultaneously, in contrast to greedy approaches that search for the optimal parameters only locally. The term is also used this way in the literature, but we understand that this might be misleading in the context of optimization techniques, which is why we clarified this in the revised version of our paper. We again want to emphasize that we do not claim that our approach finds the global optimum of the optimization problem to solve, but we argue that a sufficiently good local optimum can already outperform the sub-optimal greedy procedure.

---

### Official Review · Reviewer_zziu · 2022-10-25

**Confidence:** 4
**Correctness:** 3
**Technical Novelty And Significance:** 2
**Empirical Novelty And Significance:** 2
**Recommendation:** 5

**Clarity, Quality, Novelty And Reproducibility:**

- Clarity: The paper is clear and well-written.

- Novelty: The novelty of the approach is not clear due to missing discussion on related works as noted above. In particular the claim that this is the first approach that attempts to learn univariate, axis-aligned decision trees using gradient descent seems to contradict the results of papers [1] and [2]. [1] experimented with gumbel-softmax as well as sparse softmax variants that can become univariate/one-hot depending on their sparsity hyper-parameter and [2] learned decision trees that are univariate and axis-aligned due to the temperature annealing procedure (they use oblivious trees, but this can similarly work for non-oblivious trees).

- Reproducibility: I did not find important missing details in the paper (except for my question on the handling of categorical variables). The authors have attached their implementation and evaluation code.


**Strength And Weaknesses:**

Strengths:
* Interesting approach for computing interpretable, axis-aligned decision trees using gradient descent
* Experimental evaluation show the computed trees outperform decision trees computed by baselines in several binary classification benchmarks and are obtain competitive results on multi-class benchmarks.

Weaknesses:
* The main weakness is that the paper is missing several highly-related works on gradient-based decision trees. Several recent examples are [1], [2], [3]. While they are not identical to the propose approach (e.g., some focus on ensemble models), they all deal with computing decision trees using gradient descent algorithms. In particular, these approaches have considered similar ideas such as representing cuts as (relaxed) one-hot encoding (e.g., using sparse softmax variants in [1] [2]), computing membership of sample in a leaf using multiplication of logistic-based split functions, etc. In particular, [2] presents approach for temperature annealing that gradually turns the splits to one-hot (i.e., axis-aligned). These papers should be cited and discussed, the similarities and differences should be highlighted, and the novelty should be demonstrated compared to these approach.
* Baselines: The papers mentioned above (specifically the computing of decision trees in these papers) represent natural baselines that should be compared to in the experimental evaluation.
* The approach demonstrates relatively modest performance gain is in binary classification and underperforms compared to CART on multi-class datasaets.

Minor question: The Adult dataset has many categorical variables. How were they encoded?


[1] Popov, Sergei, Stanislav Morozov, and Artem Babenko. "Neural Oblivious Decision Ensembles for Deep Learning on Tabular Data." International Conference on Learning Representations. 2019.

[2] Chang, Chun-Hao, Rich Caruana, and Anna Goldenberg. "NODE-GAM: Neural Generalized Additive Model for Interpretable Deep Learning." International Conference on Learning Representations. 2021.

[3] Blanquero, Rafael, et al. "Sparsity in optimal randomized classification trees." European Journal of Operational Research 284.1 (2020): 255-272.


**Summary Of The Paper:**

The paper presents a new approach for computing interpretable, axis-aligned decision trees using gradient descent by applying backpropagation with adjusted gradient flow on dense representation of decision trees. Experimental evaluation finds improvement over two baselines (CART, GeneticTree) in binary classification datasets and competitive performance on multi-class datasets.

**Summary Of The Review:**

The paper presents an interesting approach for computing interpretable, axis-aligned decision trees using gradient descent. The main weaknesses are the missing highly-related literature, the limited novelty, and the limited performance gains.

---

> ### Author Response · Authors · 2022-11-18
> **Responses to your constructive feedback**
>
>
> Thank you for your constructive feedback and especially for highlighting related work that was missing in the submitted version of our paper. We addressed the related work in the revised version of our paper, and discussed the differences and novelty of our approach. A markup version highlighting the relevant text changes can additionally be found in the supplementary material. We further added the two most closely related approaches (Deep Neural Decision Trees and an Optimal Decision Tree) as benchmarks to our empirical evaluation. We were able to show that we outperform both additional benchmarks, especially on binary classification tasks.
>
> Due to lack of space, NODE and NODE-GAM could only be discussed shortly in the paper. Therefore, we wanted to address you concerns more thoroughly here:
> NODE generates ensembles of oblivious trees with end-to-end gradient-based optimization, making them related to our approach. However, oblivious trees use the same splitting feature and threshold in all internal nodes at the same depth, which makes them only suitable as weak learners for an ensemble. As mentioned by the authors, an extension of their approach to non-oblivious trees is non-trivial since it requires evaluation of d splits sequentially. In contrast, the routing implemented by our approach does not require a sequential evaluation but allows for the formulation of the tree pass function as a joint matrix operation for all nodes and all samples. Further, they utilize an entmax transformation for the choice function to select the feature(s) considered at the splits. Using the temperature annealing proposed by NODE-GAM, the split can degenerate to hard splits, making them comparable to our approach. Entmax could also be applied in our framework, but a detailed evaluation is out of scope for this paper. In general, the endmoid used as split function in both papers does not ensure hard splits, as it is the case for our approach.
>
> > The Adult dataset has many categorical variables. How were they encoded?
>
> We decided to use an ordinal encoding for all non-continuous features for simplicity and because we wanted to make as little subjective design choices as possible during preprocessing.

---

> > ### Comment · Reviewer_zziu · 2022-12-11
> > **Authors response**
> >
> > Thank you for your response.
> >
> > I think the revised paper is better and have therefore increased my evaluation score. However, my concerns regarding existing works (novelty and baselines) are not fully addressed. NODE can be easily adapted to non-oblivious trees and trained with temperature annealing. I could not find any indication in the original paper that this is non-trivial, only that inference will be less efficient compared to oblivious trees (which is not relevant in the context of the proposed approach that is also non-oblivious).

---

### Official Review · Reviewer_Dpk7 · 2022-11-03

**Confidence:** 4
**Correctness:** 2
**Technical Novelty And Significance:** 3
**Empirical Novelty And Significance:** 2
**Recommendation:** 3

**Clarity, Quality, Novelty And Reproducibility:**

The problem of greedy decision tree growth is well-known. Although there were several attempts to get rid of standard greedy procedures in the previous literature, the method prosed by the authors is novel and interesting. However, this is not the first attempt to construct a differentiable axis-aligned tree as stated in the paper (see, e.g., Yang Y. et al. Deep neural decision trees); the significance of the results is arguable.

**Strength And Weaknesses:**

+ Interesting direction and ideas

- Much work to do, the profit is questionable. According to Tavle 1, the proposed method does not outperform classical baseline CART. This is not consistent whit the claims in Introduction.

**Summary Of The Paper:**

The paper proposes a novel training algorithm for axis-aligned decision trees based on gradient descent. The tree is represented via three vectors: the first one is responsible for storing indices of nodes' split features, the second one responds the split value, and the third one stores leaf values. These vectors are represented in a contiguous form (in particular, a probability distribution for each index value) which can be further used for joint gradient descent optimization. The experimental results demonstrate comparable quality for binary classification tasks. It is highly debatable whether it is reasonable to apply the proposed method to multiclass problems.

**Summary Of The Review:**

I like the direction of the proposed method, most of the ideas are sound and interesting. However, and I think there is still much work to do.

- The statement "this is the first approach that attempts to learn univariate, axis-aligned decision trees with gradient descent" is too loud since several papers are solving the same task.

- The paper lacks theoretical justification. It would be interesting to see under what conditions one can find a global optimum and examples of synthetic data on which this method will work better than greedy construction. In particular, is this model capable of solving the "xor problem"? I lacked intuition about what the solution to the optimization problem converges to before applying hardmax, and this aspect could be discussed more. For example, it seems that if GDT is used for a regression problem with MSE loss, then the solution should converge to hard splits (but not necessarily optimal) with mean values in the leaves.

- I do not fully understand the trick described by Equation 7. Can it be interpreted as "we do not care about soft and contiguous nature of \iota and S during training but apply hard max at the end"? If so, I do not see any problems with it; it could be written explicitly for more clarity.

- The choice of baseline algorithms is not clear. Why were Evolutionary Decision Trees used but not some other differentiable trees? Results show that it is quite a weak baseline, even compared to classic CART trees.

- The experimental part would be more solid if it consisted of examples of surfaces found by greedy tree and GDT and the plot demonstrating quality vs cardinality dependence.

- I do not understand why potentially more expressive GDT is less prone to overfitting and, at the same time, has comparable or even lower quality. It can be due to the validation set used for GDT training. Other algorithms do not use them, but many pruning techniques could make them more robust, and at the same time, reducing the train set may be crucial for your model (especially with small datasets).

* Questions

- Unfortunately, the results are not so good for some datasets. What can you state about the limitations of applying the proposed method to real-world problems (how to decide whether to use it or not)?

- Experience shows that using more expressive trees in ensembles (such as GBDT, for example) can lead to a decrease in the quality of such models due to overfitting on the level of individual trees. Have you tried using GDT in ensembles? Does it make sense, especially in the context of much longer training time?

---

> ### Author Response · Authors · 2022-11-18
> **Response to your helpful feedback**
>
> We thank the reviewer for their useful and constructive feedback. Many details are already addressed in the revised version of the paper which we have uploaded. A markup version highlighting the relevant text changes can additionally be found in the supplementary material. Most importantly, we added DNDT and DL8.5 to our empirical evaluation, which were helpfully pointed out by reviewers as relevant benchmark methods. Please see our responses addressing the specific concerns below:
>
> > The paper lacks theoretical justification. It would be interesting to see under what conditions one can find a global optimum
>
> With the expression “global optimization” we wanted to emphasize that our gradient-based approach optimizes all tree parameters simultaneously, in contrast to greedy approaches that search for the optimal parameters only locally. The term is also used this way in the literature, but we understand that this might be misleading in the context of optimization techniques, which is why we clarified this in the revised version of our paper and adjusted the formulation to “joint optimization of all tree parameters”. We again want to emphasize that we do not claim that our approach finds the global optimum of the optimization problem. However, we argue that a sufficiently good local optimum can already outperform a sub-optimal greedy procedure.
>
> > The statement "this is the first approach that attempts to learn univariate, axis-aligned decision trees with gradient descent" is too loud since several papers are solving the same task.
>
> We acknowledge the prior work on differentiable decision trees and therefore adjusted the formulation in the revised version of the paper as follows: “this is the first approach that attempts to learn hard, axis-aligned decision trees with gradient descent without restrictions regarding its structure”. We also discussed this more thoroughly in the related work of the revised version, and differentiated our approach from existing work on gradient-based DTs.
>
> > The choice of baseline algorithms is not clear. Why were Evolutionary Decision Trees used but not some other differentiable trees? Results show that it is quite a weak baseline, even compared to classic CART trees.
>
> Most differentiable trees proposed in the literature are either soft and/or oblique, which is not an appropriate benchmark for our hard, axis-aligned DTs. However, in the revised version, we discussed related differentiable approaches more thoroughly. In addition, we added Deep Neural Decision Trees (DNDT) and an Optimal Decision Tree (DL8.5) to the evaluation. We were able to show that GDTs outperform both new baselines, especially on binary classification tasks.
>
> > I do not fully understand the trick described by Equation 7. Can it be interpreted as "we do not care about soft and contiguous nature of \iota and S during training but apply hard max at the end"? If so, I do not see any problems with it; it could be written explicitly for more clarity.
>
> Yes, more specifically, as noted by reviewer ePjz, we use the straight-trough operator to deal with the non-differentiable nature of the choice function (application of hardmax to \iota) and the split function (S). We already clarified this in the revised version of the paper and stated it more explicitly.
>
> > I do not understand why potentially more expressive GDT is less prone to overfitting and, at the same time, has comparable or even lower quality.
>
> We measure overfitting by the difference between the performance on the training data and the test data. If the performance on the training data is significantly higher, this is a sign of overfitting. During our experiments, we observed that greedy DT algorithms tend to have a significantly higher performance on the training data. We can trace this back to the fact that the purity-based selection of the locally optimal split frequently leads to splits that only separate a small fraction of samples where a high purity can be achieved. We are aware that this can be addressed by regularization (e.g., using the min_samples_leaf parameter), however we observed that this comes at the cost of a performance loss. The regularization parameters were also considered during the cross-validated hyperparameter optimization.
> Empirically, we observe a lower difference between test and train accuracy in GDT than in other DT learners. Intuitively, one explanation might be the fact that a purity-based selection of the locally optimal split frequently leads to splits that only separate a small fraction of samples where a high purity can be achieved, while GDTs always consider the global performance.

---

> ### Author Response · Authors · 2022-11-18
> **Response to your open questions**
>
>
> > Unfortunately, the results are not so good for some datasets. What can you state about the limitations of applying the proposed method to real-world problems (how to decide whether to use it or not)?
>
> We are aware that we are only able to outperform CART on several datasets. Therefore, we believe that including our approach in the model selection step of the ML pipeline and deciding on the model to use based on cross validation is the most reliable procedure. In general, our approach excels on binary classification tasks with a small to moderate number of features (<15).
>
> > Experience shows that using more expressive trees in ensembles (such as GBDT, for example) can lead to a decrease in the quality of such models due to overfitting on the level of individual trees. Have you tried using GDT in ensembles? Does it make sense, especially in the context of much longer training time?
>
> We have not yet evaluated GDTs in ensembles but want to address this in future work. In general, we believe that GDTs are well-suited to be used in ensembles, since they are very robust individually. However, we think that it is reasonable to apply regularization/pruning to GDTs first. Regarding the training time we do not believe that problems will arise for two reasons: (1) The runtime of GDTs for lower depths, which is commonly used in ensembles, is very small (only a few seconds), even for large datasets and on CPU. (2) The training of the GDTs in an ensemble can be easily parallelized.

---

### Comment · Area_Chair_yTQv · 2022-12-07
**Response to Author Feedback**

Dear Reviewers, thank you so much again for your time on this paper. The discussion phase is still ongoing, how does the author response and other reviews change your view of the paper?

---

### Decision · Program_Chairs · 2023-01-20

**Decision:**

Reject

**Justification For Why Not Higher Score:**

See above metareview.

**Justification For Why Not Lower Score:**

N/A

**Metareview: Summary, Strengths And Weaknesses:**

While the reviewers appreciated the paper’s clear contributions, the novelty of the idea, and the clarity of the paper, their main concerns were with (a) the motivation for learning axis-aligned splits, and (b) the description of related work. Specifically, for (a) the authors argue that trees with axis-aligned splits are more interpretable than oblique trees. However, the reviewers pointed out that oblique trees could be much shallower than axis-aligned ones. These fewer decision rules could be easier to interpret. The proposed method learns fully-grown trees which are often considered less interpretable e.g., see Rudin et al., 2022. The authors partially concede this point, and they revise their motivation argument as follows: for high-dimensional problems shallow oblique trees are less interpretable than axis-aligned trees at any depth because, they say, according to Molnar, 2020, “humans can't comprehend explanations considering more than three features at the time”. However, the reviewer points out that this is no longer an argument about axis-aligned trees vs. oblique trees: but about the number of required decision functions. As the proposed method learns fully-grown trees, it’s entirely possible that the number of decision trees exceeds three decisions, and potentially the number of decisions of a shallow oblique tree (as oblique trees can also use a sparse set of features). For (b), the authors respond by including the suggested related work and pointing out that optimal decision tree methods methods require binary features. However, this is not true, e.g., Zhu et al., A Scalable MIP-based Method for Learning Optimal Multivariate Decision Trees, 2020. Given these open points, I believe this work should be rejected at this time. Once these things are clarified this paper will be improved.